# Inhibitory muscarinic acetylcholine receptors enhance aversive olfactory learning in adult *Drosophila*

Noa Bielopolski[1], Hoger Amin[2†], Anthi A Apostolopoulou[2†], Eyal Rozenfeld[1†], Hadas Lerner[1], Wolf Huetteroth[3], Andrew C Lin[2‡*], Moshe Parnas[1,4‡*]

[1]Department of Physiology and Pharmacology, Sackler School of Medicine, Tel Aviv University, Tel Aviv, Israel; [2]Department of Biomedical Science, University of Sheffield, Sheffield, United Kingdom; [3]Institute for Biology, University of Leipzig, Leipzig, Germany; [4]Sagol School of Neuroscience, Tel Aviv University, Tel Aviv, Israel

**Abstract** Olfactory associative learning in *Drosophila* is mediated by synaptic plasticity between the Kenyon cells of the mushroom body and their output neurons. Both Kenyon cells and their inputs from projection neurons are cholinergic, yet little is known about the physiological function of muscarinic acetylcholine receptors in learning in adult flies. Here, we show that aversive olfactory learning in adult flies requires type A muscarinic acetylcholine receptors (mAChR-A), particularly in the gamma subtype of Kenyon cells. mAChR-A inhibits odor responses and is localized in Kenyon cell dendrites. Moreover, mAChR-A knockdown impairs the learning-associated depression of odor responses in a mushroom body output neuron. Our results suggest that mAChR-A function in Kenyon cell dendrites is required for synaptic plasticity between Kenyon cells and their output neurons.
DOI: https://doi.org/10.7554/eLife.48264.001

**\*For correspondence:**
andrew.lin@sheffield.ac.uk (ACL);
mparnas@tauex.tau.ac.il (MP)

[†]These authors contributed equally to this work
[‡]These authors also contributed equally to this work

**Competing interests:** The authors declare that no competing interests exist.

## Introduction

Animals learn to modify their behavior based on past experience by changing connection strengths between neurons, and this synaptic plasticity is often regulated by metabotropic receptors. In particular, neurons commonly express both ionotropic and metabotropic receptors for the same neurotransmitter, where the two may mediate different functions (e.g. direct excitation/inhibition vs. synaptic plasticity). In mammals, where glutamate is the principal excitatory neurotransmitter, metabotropic glutamate receptors (mGluRs) have been widely implicated in synaptic plasticity and memory (*Jörntell and Hansel, 2006*; *Lüscher and Huber, 2010*). Given the complexity of linking behavior to artificially induced plasticity in brain slices (*Schonewille et al., 2011*; *Yamaguchi et al., 2016*), it would be useful to study the role of metabotropic receptors in learning in a simpler genetic model system with a clearer behavioral readout of synaptic plasticity. One such system is *Drosophila*, where powerful genetic tools and well-defined anatomy have yielded a detailed understanding of the circuit and molecular mechanisms underlying associative memory (*Busto et al., 2010*; *Cognigni et al., 2018*; *Hige, 2018*). The principal excitatory neurotransmitter in *Drosophila* is acetylcholine, but, surprisingly, little is known about the function of metabotropic acetylcholine signaling in synaptic plasticity or neuromodulation in *Drosophila*. Here, we address this question using olfactory associative memory.

Flies can learn to associate an odor (conditioned stimulus, CS) with a positive (sugar) or a negative (electric shock) unconditioned stimulus (US), so that they later approach 'rewarded' odors and avoid 'punished' odors. This association is thought to be formed in the presynaptic terminals of the ~2000

**eLife digest** We can learn a surprising amount about how the brain forms memories by studying the humble fruit fly. These insects can learn to associate odors with positive or negative experiences, allowing them to then seek out 'rewarded' odors and avoid 'punished' ones.

This association takes place in a brain region called the mushroom body, and it involves two types of neurons: Kenyon cells, which detect odors, and MBONs, which lead to approach or avoidance behaviors. When Kenyon cells detect an odor accompanying an unpleasant event, they weaken their connections with the MBONs that trigger approach behaviors. This prevents the fly from coming close to that odor in the future.

Kenyon cells exchange signals with other neurons using a chemical called acetylcholine, which attaches onto the cells through two types of receptors: nicotinic and muscarinic. Studies in fruit fly larvae suggest that muscarinic receptors are required in Kenyon cells for the insects to learn how to associate odors with unpleasant experiences.

Bielopolski et al. now show that this is also the case in adult flies. Surprisingly, while acetylcholine usually excites fly neurons, activating muscarinic receptors inhibits Kenyon cells rather than exciting them. Labeled muscarinic receptors revealed that the receptors act within the input region of Kenyon cells. Moreover, reducing the levels of muscarinic receptors inside the cells stops flies from associating an odor with a mild electric shock. This manipulation also prevents the learning experience from weakening connections from Kenyon cells onto an MBON that triggers approach behavior. This suggests that allowing these changes in connectivity might be why muscarinic receptors are important for memory.

Understanding how memory works in flies can reveal basic principles that apply to many species, including humans. Such knowledge could ultimately help us improve the memory of patients with dementia, but also inspire better algorithms for artificial intelligence.

DOI: https://doi.org/10.7554/eLife.48264.002

Kenyon cells (KCs) that make up the mushroom body (MB), the fly's olfactory memory center (*Busto et al., 2010*; *Cognigni et al., 2018*; *Hige, 2018*). These KCs are activated by odors via second-order olfactory neurons called projection neurons (PNs). Each odor elicits responses in a sparse subset of KCs (*Campbell et al., 2013*; *Lin et al., 2014*) so that odor identity is encoded in which KCs respond to each odor. When an odor (CS) is paired with reward/punishment (US), an odor-specific set of KCs is activated at the same time that dopaminergic neurons (DANs) release dopamine onto KC presynaptic terminals. The coincident activation causes long-term depression (LTD) of synapses from the odor-activated KCs onto mushroom body output neurons (MBONs) that lead to approach or avoidance behavior (*Aso and Rubin, 2016*; *Aso et al., 2014b*; *Cohn et al., 2015*; *Hige et al., 2015*; *Owald et al., 2015*; *Perisse et al., 2016*; *Séjourné et al., 2011*). In particular, training specifically depresses KC-MBON synapses of the 'wrong' valence (e.g. odor-punishment pairing depresses odor responses of MBONs that lead to approach behavior), because different pairs of 'matching' DANs/MBONs (e.g. punishment/approach, reward/avoidance) innervate distinct regions along KC axons (*Aso et al., 2014a*).

Both MB input (PNs) and output (KCs) are cholinergic (*Barnstedt et al., 2016*; *Yasuyama and Salvaterra, 1999*), and KCs express both ionotropic (nicotinic) and metabotropic (muscarinic) acetylcholine receptors (*Crocker et al., 2016*; *Croset et al., 2018*; *Davie et al., 2018*; *Shih et al., 2019*). The nicotinic receptors mediate fast excitatory synaptic currents (*Su and O'Dowd, 2003*), while the physiological function of the muscarinic receptors is unknown. Muscarinic acetylcholine receptors (mAChRs) are G-protein-coupled receptors; out of the three mAChRs in *Drosophila* (mAChR-A, mAChR-B and mAChR-C), mAChR-A (also called Dm1, mAcR-60C or mAChR) is the most closely homologous to mammalian mAChRs (*Collin et al., 2013*). Mammalian mAChRs are typically divided between 'M$_1$-type' (M$_1$/M$_3$/M$_5$), which signal via G$_q$ and are generally excitatory, and 'M$_2$-type' (M$_2$/M$_4$), which signal via G$_{i/o}$ and are generally inhibitory (*Caulfield and Birdsall, 1998*). *Drosophila* mAChR-A seems to use 'M$_1$-type' signaling: when heterologously expressed in Chinese hamster ovary (CHO) cells, it signals via G$_q$ protein (*Collin et al., 2013*; *Ren et al., 2015*) to activate phospholipase C, which produces inositol trisphosphate to release Ca$^{2+}$ from internal stores.

Recent work indicates that mAChR-A is required for aversive olfactory learning in *Drosophila* larvae, as knocking down mAChR-A expression in KCs impairs learning (*Silva et al., 2015*). However, it is unclear whether mAChR-A is involved in olfactory learning in adult *Drosophila*, given that mAChR-A is thought to signal through $G_q$, and in adult flies $G_q$ signaling downstream of the dopamine receptor Damb promotes forgetting, not learning (*Berry et al., 2012*; *Himmelreich et al., 2017*). Moreover, it is unknown how mAChR-A affects the activity or physiology of KCs, where it acts (at KC axons or dendrites or both), and how these effects contribute to olfactory learning.

Here, we show that mAChR-A is required in KCs for aversive olfactory learning in adult *Drosophila*. Surprisingly, genetic and pharmacological manipulations of mAChR-A suggest that mAChR-A is inhibitory and acts on KC dendrites. Moreover, mAChR-A knockdown impairs the learning-associated depression of odor responses in an MB output neuron, MB-MVP2, that is required for aversive memory retrieval. We suggest that dendritically acting mAChR-A is required for synaptic depression between KCs and their outputs.

## Results

### mAChR-A expression in KCs is required for aversive olfactory learning in adult flies

*Drosophila* larvae with reduced mAChR-A expression in KCs show impaired aversive olfactory learning (*Silva et al., 2015*), but it remains unknown whether mAChR-A in KCs also functions in learning in adult flies. We addressed this question by knocking down mAChR-A expression in KCs using two UAS-RNAi lines, 'RNAi 1' and 'RNAi 2' (see Materials and methods). Only RNAi 2 requires co-expression of Dicer-2 (Dcr-2) for optimal knockdown. To test the efficiency of these RNAi constructs, we expressed them pan-neuronally using elav-GAL4 and measured their effects on mAChR-A expression levels using quantitative real-time polymerase chain reaction (qRT-PCR). Both RNAi lines strongly reduce mAChR-A levels (RNAi 1: 39 ± 8% of elav-GAL4 control, or 61 ± 8% below normal; RNAi 2: 43 ± 10% of normal; mean ± s.e.m.; see *Figure 1A*). We then examined whether knocking down mAChR-A in KCs using the pan-KC driver OK107-GAL4 affects short-term aversive learning in adult flies. We used the standard odors used in the field (i.e. 3-octanol, OCT, and 4-methylcyclohexanol, MCH; see Materials and methods). Under these conditions, both UAS-RNAi transgenes significantly reduced aversive learning, whether training against MCH or OCT (*Figure 1B,C* and *Figure 1—figure supplement 1*). Interestingly, knocking down mAChR-A did not affect learning when we trained flies with a more intense shock (90 V instead of 50 V, *Figure 1—figure supplement 1*), suggesting that mAChR-A may only be required for learning with moderate intensity reinforcement, not severe reinforcement. Consistent with this, knocking down mAChR-A had no effect on naïve avoidance of MCH and OCT (*Figure 1D*; see Materials and methods) or flies' reaction to electric shock (*Figure 1—figure supplement 1*), showing that the defect was specific to learning, rather than reflecting a failure to detect odors or shock.

Given that mAChR-A is expressed in the larval MB and indeed contributes to aversive learning in larvae, it is possible that developmental effects underlie the reduced learning observed in mAChR-A knockdown flies. To test this, we used tub-GAL80$^{ts}$ to suppress RNAi 1 expression during development. Flies were grown at 23°C until 3 days after eclosion and were then transferred to 31°C for 7 days. Adult-only knockdown of mAChR-A in KCs reduced learning (*Figure 1E*), just as constitutive knockdown did, indicating that mAChR-A plays a physiological, not purely developmental, role in aversive learning. To further verify that GAL80$^{ts}$ efficiently blocks RNAi expression (i.e. that GAL80$^{ts}$ is not leaky), flies were grown at 23°C without transferring them to 31°C, thus blocking RNAi expression also in adults. When tested for learning at 10 days old, these flies showed normal learning (*Figure 1E*).

### mAChR-A is required for olfactory learning in γ KCs, not αβ or α'β' KCs

Kenyon cells are subdivided into three main classes according to their innervation of the horizontal and vertical lobes of the MB: γ neurons send axons only to the γ lobe of the horizontal lobes, while the axons of αβ and α'β' neurons bifurcate and go to both the vertical and horizontal lobes (αβ axons make up the α lobe of the vertical lobe and β lobe of the horizontal lobe, while α'β' axons make up the α' lobe of the vertical lobe and β' portion of the horizontal lobe). These different classes

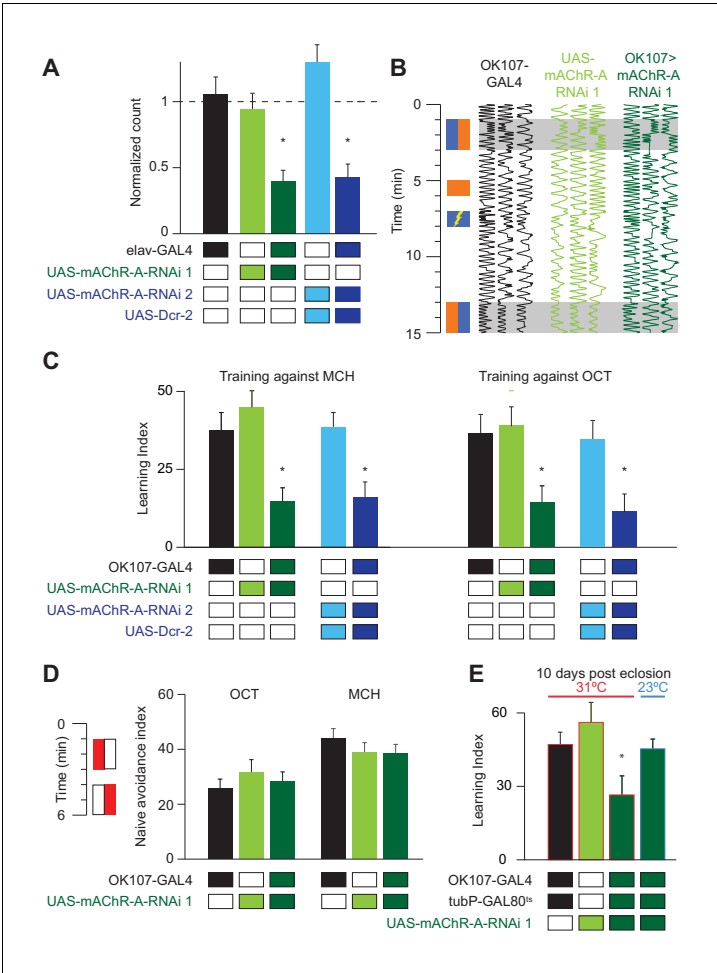

**Figure 1.** mAChR-A is required in the MB for short-term aversive olfactory learning and memory but not for naive behavior. (A) qRT-PCR of mAChR-A with mAChR-A RNAi driven by elav-GAL4. The housekeeping gene eEF1α2 (eukaryotic translation elongation factor 1 alpha 2, CG1873) was used for normalization. Knockdown flies have ~40% of the control levels of mAChR-A mRNA (mean ± SEM; number of biological replicates (left to right): 6, 7, 7, 4, 4, each with three technical replicates; *p<0.05; Kruskal-Wallis test with Dunn's multiple comparisons test and Welch ANOVA test with Dunnett's T3 multiple comparisons test). For detailed statistical analysis see *Supplementary file 1*. (B) Each trace shows the movement of an individual fly during the training protocol, with fly position in the chamber (horizontal dimension) plotted against time (vertical dimension). Colored rectangles illustrate which odor is presented on each side of the chamber during training and testing. Flies were conditioned against MCH (blue rectangles; see Materials and methods). (C) Learning scores in flies with mAChR-A RNAi driven by OK107-GAL4. mAChR-A knockdown reduced learning scores compared to controls (mean ± SEM, n (left to right): 69, 69, 70, 71, 71, 47, 48, 53, 58, 51 *p<0.05; Kruskal-Wallis test with Dunn's multiple comparisons test). (D) mAChR-A knockdown flies show normal olfactory avoidance to OCT and MCH compared to their genotypic controls (mean ± SEM, n (left to right): 68, 67, 58, 63, 91, 67, p=0.82 for OCT, p=0.64 for MCH; Kruskal-Wallis test). Colored rectangles show stimulus protocol as in (B); red for odor (MCH or OCT), white for air. (E) Learning scores in flies with mAChR-A RNAi 1 driven by OK107-GAL4 with GAL80ts repression. Flies raised at 23°C and heated to 31°C as adults (red outlines) had impaired learning compared to controls. Control flies kept at 23°C throughout (blue outline), thus blocking mAChR-A RNAi expression, showed no learning defects (mean ± SEM, n (left to right): 51, 41, 58, 51, **p<0.05, Kruskal-Wallis test with Dunn's multiple comparisons test). For detailed statistical analysis see *Supplementary file 1*.

DOI: https://doi.org/10.7554/eLife.48264.003

The following source data and figure supplements are available for figure 1:

**Source data 1.** Source data for *Figure 1A*.
DOI: https://doi.org/10.7554/eLife.48264.006

**Source data 2.** Source data for *Figure 1C–E*.
DOI: https://doi.org/10.7554/eLife.48264.007

**Figure supplement 1.** Controls and additional learning data.
DOI: https://doi.org/10.7554/eLife.48264.004

**Figure supplement 1—source data 1.** Source data for *Figure 1—figure supplement 1*.
DOI: https://doi.org/10.7554/eLife.48264.005

play different roles in olfactory learning (*Guven-Ozkan and Davis, 2014*; *Krashes et al., 2007*). To unravel in which class(es) mAChR-A functions, we used a Minos-mediated integration cassette (MiMIC) line to investigate where mAChR-A is expressed (*Venken et al., 2011*). The MiMIC insertion in mAChR-A lies in the first 5' non-coding intron, creating a gene trap where GFP in the MiMIC cassette should be expressed in whichever cells endogenously express mAChR-A. Because the GFP in the original mAChR-A MiMIC cassette produced very little fluorescent signal (data not shown), we used recombinase-mediated cassette exchange (RMCE) to replace the original MiMIC cassette with a MiMIC cassette containing GAL4 (*Venken et al., 2011*). These new mAChR-A-MiMIC-GAL4 flies should express GAL4 wherever mAChR-A is endogenously expressed. To reveal the expression pattern of mAChR-A, we crossed mAChR-A-MiMIC-GAL4 and 20xUAS-6xeGFP flies. mAChR-A-MiMIC-GAL4 drove GFP expression throughout the brain, consistent with previous reports (*Blake et al., 1993*; *Croset et al., 2018*; *Davie et al., 2018*; *Hannan and Hall, 1996*) and with the fact that the *Drosophila* brain is mostly cholinergic. In the mushroom bodies, GFP was expressed in the αβ and γ lobes, but not the α'β' lobes (*Figure 2A*). No GFP signal was observed with an inverted insertion where GAL4 is inserted in the MiMIC locus in the wrong orientation (data not shown). Consistent with these MiMIC results, two recently reported databases of single-cell transcriptomic analysis of the *Drosophila* brain (*Croset et al., 2018*; *Davie et al., 2018*) confirm that mAChR-A is more highly expressed in αβ and γ KCs than in α'β' KCs (*Figure 2—figure supplement 1*). However, mAChR-A is still clearly present in α'β' KCs' transcriptomes, suggesting that mAChR-A-MiMIC-GAL4 may not reveal all neurons that express mAChR-A.

The higher expression of mAChR-A in αβ and γ KCs compared to α'β' KCs suggests that learning would be impaired by mAChR-A knockdown in αβ or γ, but not α'β', KCs. To test this, we expressed mAChR-A RNAi in different KC classes. As expected, aversive olfactory learning was reduced by

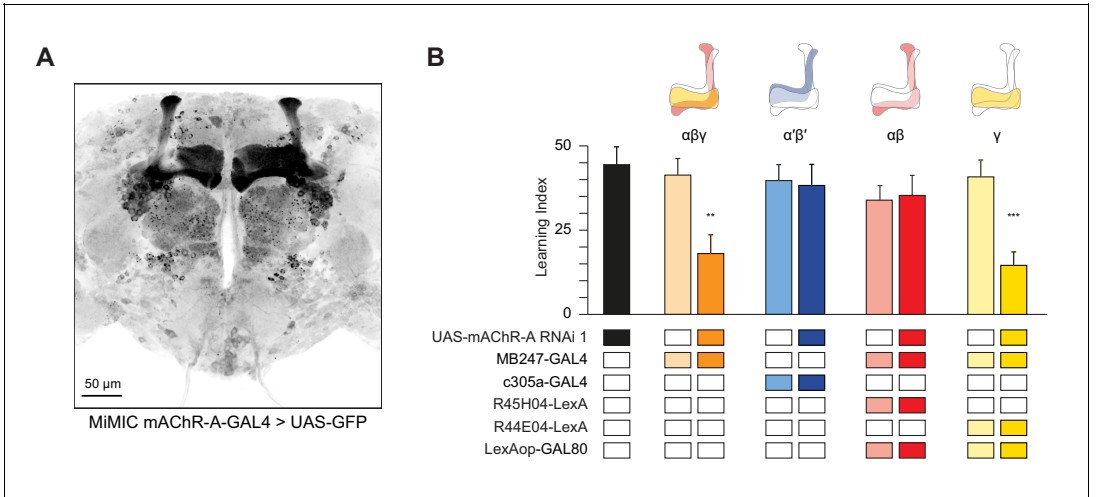

**Figure 2.** mAChR-A is required for short-term aversive olfactory learning and memory in γ KCs. (**A**) Maximum intensity projection of 70 confocal sections (2 μm) through the central brain of a fly carrying MiMIC-mAChR-A-GAL4 and 20xUAS-6xGFP transgenes. MB αβ and γ lobes are clearly observed. No GFP expression is observed in α'β' lobes. (**B**) mAChR-A RNAi 1 was targeted to different subpopulations of KCs. Learning scores were reduced compared to controls when mAChR-A RNAi 1 was expressed in αβ and γ KCs or γ KCs alone, but not when mAChR-A RNAi 1 was expressed in αβ or α'β' KCs. (mean ± SEM, n (left to right): 69, 41, 70, 76, 69, 66, 71, 50, 68, **p<0.01, ***p<0.001, Kruskal-Wallis test with Dunn's multiple comparisons test). For detailed statistical analysis see *Supplementary file 1*. The data for the UAS-mAChR-A RNAi 1 control are duplicated from *Figure 1*.

DOI: https://doi.org/10.7554/eLife.48264.008

The following source data and figure supplements are available for figure 2:

**Source data 1.** Source data for *Figure 2*.
DOI: https://doi.org/10.7554/eLife.48264.011
**Figure supplement 1.** Expression of mAChR-A from single-cell transcriptome profiling.
DOI: https://doi.org/10.7554/eLife.48264.009
**Figure supplement 2.** Expression patterns of GAL4 and LexA driver lines used in this study.
DOI: https://doi.org/10.7554/eLife.48264.010

knocking down mAChR-A in αβ and γ KCs together using MB247-GAL4, but not by knockdown in α′β′ KCs using c305a-GAL4. To examine if αβ and γ KCs both participate in the reduced learning observed in mAChR-A knockdown flies, we sought to limit mAChR-A RNAi expression to either αβ or γ neurons. While strong driver lines exist for αβ neurons, the γ GAL4 drivers we tested were fairly weak (H24-GAL4, MB131B, R45H04-GAL4, data not shown), perhaps too weak to drive mAChR-A-RNAi enough to knock down mAChR-A efficiently. Therefore, we used MB247-GAL4, which was strong enough to affect behavior, and blocked GAL4 activity in either αβ or γ KCs by expressing the GAL80 repressor under the control of R44E04-LexA (αβ KCs) or R45H04-LexA (γ KCs) (*Bräcker et al., 2013*). These combinations drove strong, specific expression in αβ or γ KCs (*Figure 2—figure supplement 2*). Learning was reduced by mAChR-A RNAi expression in γ, but not αβ, KCs (*Figure 2B*). These results suggest that mAChR-A is specifically required in γ KCs for aversive olfactory learning and short-term memory.

## mAChR-A suppresses odor responses in γ KCs

We next asked what effect mAChR-A knockdown has on the physiology of KCs, by expressing GCaMP6f and mAChR-A RNAi 2 together in KCs using OK107-GAL4 (this driver and RNAi combination was also used for behavior in *Figure 1C*). Knocking down mAChR-A in KCs increased odor-evoked $Ca^{2+}$ influx in the mushroom body calyx, where KC dendrites reside (*Figure 3*). This result is somewhat surprising because mAChR-A is a $G_q$-coupled receptor whose activation leads to $Ca^{2+}$ release from internal stores (*Ren et al., 2015*), which predicts that mAChR-A knockdown should decrease, not increase, odor-evoked $Ca^{2+}$ influx in KCs. However, some examples have been reported of inhibitory signaling through $G_q$ by $M_1$-type mAChRs (see Discussion), and *Drosophila* mAChR-A may join these as another example of an inhibitory mAChR signaling through $G_q$.

Because mAChR-A is required for aversive learning in γ KCs, not αβ or α′β′ KCs (*Figure 2*), we next asked how odor responses in αβ, α′β′ and γ KCs are affected by mAChR-A knockdown. αβ, α′β′ and γ KC dendrites are not clearly segregated in the calyx, so we examined odor responses in the axonal lobes. Indeed, although odor responses in all lobes were increased by mAChR-A knockdown, only in the γ lobe was the effect statistically significant for both MCH and OCT (*Figure 3*). This result is consistent with the behavioral requirement for mAChR-A only in γ KCs. However, we do not rule out the possibility that mAChR-A knockdown also affects αβ and α′β′ odor responses in a way that does not affect short-term memory, especially as αβ and α′β′ odor responses were somewhat, although not consistently significantly, increased. Although the ΔF/F traces from the γ lobe had higher signal-to-noise ratio (SNR) than some other lobes (*Figure 3—figure supplement 1*) due to its larger size (averaging over more pixels) or shallower z-depth (less light scattering), a power analysis revealed that all lobes had SNRs high enough to detect an effect as large as that observed in the γ lobe (*Figure 3—figure supplement 1*). However, note that we do not exclude the possibility that αβ- or α′β′-specific (as opposed to pan-KC) knockdown of mAChR-A might significantly increase αβ or α′β′ KC odor responses.

Do increased odor responses in γ KCs prevent learning by increasing the overlap between the γ KC population representations of the two odors used in our task (*Lin et al., 2014*)? When GCaMP6f and mAChR-A-RNAi 2 were expressed in all KCs, mAChR-A knockdown did not affect the sparseness or inter-odor correlation of KC population odor responses (*Figure 4A–C*) even though it increased overall calyx responses. To focus specifically on γ KCs, we expressed GCaMP6f and mAChR-A-RNAi 1 only in γ KCs, using mb247-Gal4, R44E04-LexA and lexAop-GAL80, the same driver and RNAi combination used in the behavioral experiments in *Figure 2B*. GCaMP6f was visible mainly in the γ lobe (*Figure 4D*). γ-only expression of mAChR-A-RNAi 1 increased odor responses in the calyx (here, dendrites of γ KCs only) and, in the case of OCT, in the γ lobe (*Figure 4E,F*). Note that γ KC odor responses were increased by both RNAi 1 (*Figure 3A,B*) and RNAi 2 (*Figure 4E,F*). As with pan-KC expression, γ-only expression of mAChR-A-RNAi 1 did not affect the sparseness or inter-odor correlation of γ KCs (*Figure 4G–I*). Thus, mAChR-A knockdown does not impair learning through increased overlap in KC population odor representations.

## KC odor responses are decreased by an mAChR agonist

RNAi-based knockdown of mAChR-A might induce homeostatic compensation that obscures or even reverses the primary effect of reduced mAChR-A expression. To test the acute role of

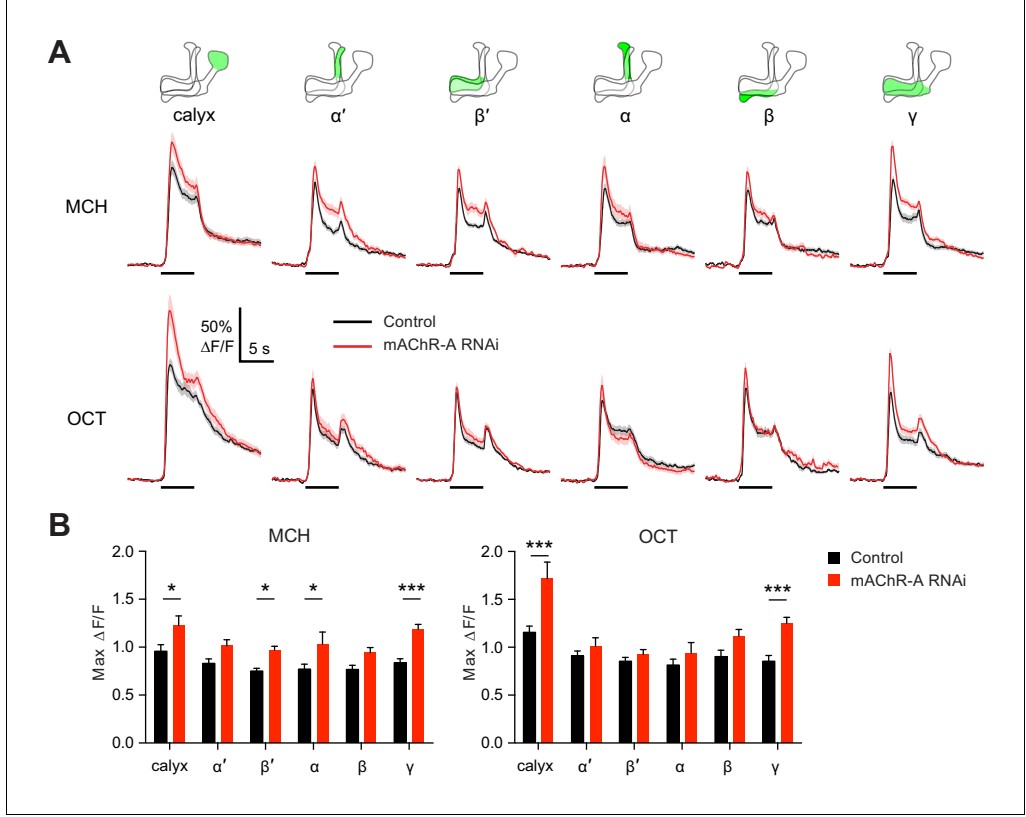

**Figure 3.** mAChR-A knockdown increases odor responses in γ KCs. Odor responses to MCH and OCT were measured in control (OK107-GAL4 > GCaMP6f, Dcr-2) and knockdown (OK107-GAL4 > GCaMP6f, Dcr-2, mAChR-A-RNAi 2) flies. (**A**) ΔF/F of GCaMP6f signal in different areas of the MB in control (black) and knockdown (red) flies, during presentation of odor pulses (horizontal lines). Data are mean (solid line) ± SEM (shaded area). Diagrams illustrate which region of the MB was analyzed. (**B**) Peak response of the traces presented in A (mean ± SEM). n given as number of hemispheres (number of flies) for control and knockdown flies, respectively: calyx, 23 (13), 17 (10); α and α', 24 (13), 20 (10); β, β' and γ, 27 (14), 22 (11). *p<0.05, ***p<0.001, two-way ANOVA with Holm-Sidak multiple comparisons test). For detailed statistical analysis see *Supplementary file 1*.
DOI: https://doi.org/10.7554/eLife.48264.012

The following source data and figure supplement are available for figure 3:

**Source data 1.** Source data for *Figure 3*.
DOI: https://doi.org/10.7554/eLife.48264.014
**Figure supplement 1.** Statistical power is not affected by inter-lobe differences in signal-to-noise ratio (SNR).
DOI: https://doi.org/10.7554/eLife.48264.013

mAChR-A in regulating KC activity, we took the complementary approach of pharmacologically activating mAChR-A. Initially, we bath-applied 10 µM muscarine, an mAChR-A agonist (*Drosophila* mAChR-B is 1000-fold less sensitive to muscarine than mAChR-A is [*Collin et al., 2013*], and mAChR-C is not expressed in the brain [*Davie et al., 2018*]). Muscarine strongly decreased odor responses in all subtypes of KCs (*Figure 5A,B*, *Figure 5—figure supplement 1*). However, muscarine did not significantly affect the amplitude of odor responses in PN axons in the calyx (*Figure 5C*), suggesting that the effect of muscarine on KCs arose in KCs, not earlier in the olfactory pathway. KCs can be silenced by an inhibitory GABAergic neuron called the anterior paired lateral (APL) neuron (*Lin et al., 2014*; *Masuda-Nakagawa et al., 2014*; *Papadopoulou et al., 2011*), so we asked whether muscarine reduces KC odor responses indirectly by activating APL, rather than directly inhibiting KCs. We applied muscarine to flies with APL-specific expression of tetanus toxin (TNT), which blocks inhibition from APL and thereby greatly increases KC odor responses. In these flies, APL is labeled stochastically, so hemispheres where APL was unlabeled served as controls (*Lin et al., 2014*) (see Materials and methods). Muscarine decreased KC odor responses both in

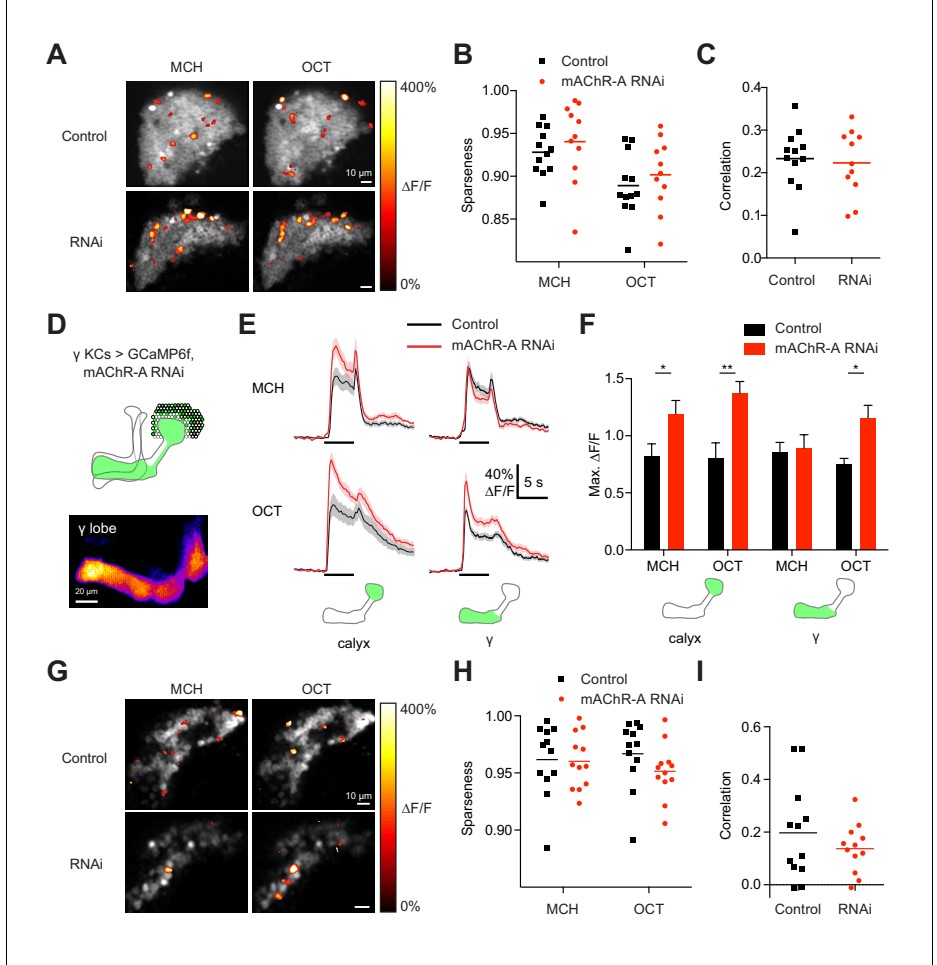

**Figure 4.** mAChR-A knockdown does not affect KC odor identity coding. (**A**) Example activity maps (single optical sections from a z-stack) of KC odor responses to MCH and OCT in control (OK107-GAL4 > GCaMP6f, Dcr-2) and mAChR-A knockdown (OK107-GAL4 > GCaMP6f, Dcr-2, mAChR-A-RNAi 2) flies where all KCs are imaged. False-coloring indicates ΔF/F of the odor response, overlaid on grayscale baseline GCaMP6f signal. Scale bar, 10 μm. For detailed statistical analysis see ***Supplementary file 1***. (**B**) Sparseness of pan-KC population responses is not affected by mAChR-A knockdown (p=0.38, two-way repeated-measures ANOVA). (**C**) Correlation between pan-KC population responses to MCH and OCT is not affected by mAChR-A knockdown (p=0.75, t-test). (**D**) Upper: diagram of γ KCs (green). Lower: False-colored average-intensity Z-projection of the horizontal lobe in a control fly imaged from a dorsal view in panel E (mb247-GAL4 > GCaMP6f, R44E04-LexA > GAL80), averaged over 10 s before the odor stimulus. R44E04-LexA > GAL80 almost completely suppresses β lobe expression. Scale bar, 20 μm. (**E**) Knocking down mAChR-A only in γ KCs increases γ KC odor responses. Shown here are odor responses in the calyx and γ lobe of control (mb247-GAL4 > GCaMP6f, R44E04-LexA > GAL80) and knockdown (mb247-GAL4 > GCaMP6f, mAChR-A-RNAi 1, R44E04-LexA > GAL80) flies. (**F**) Peak response of the traces presented in D (mean ± SEM.) n given as number of hemispheres (number of flies): 11 (6) for control, 12 (6) for knockdown. *p<0.05, **p<0.01, 2-way repeated-measures ANOVA with Holm-Sidak multiple comparisons test. (**G**) Example activity maps (single optical sections from a z-stack) of γ KC odor responses to MCH and OCT in control (mb247-GAL4 > GCaMP6f, R44E04-LexA > GAL80) and knockdown (mb247-GAL4 > GCaMP6f, mAChR-A-RNAi 1, R44E04-LexA > GAL80) flies. Note the gaps in baseline GCaMP6f signal due to lack of αβ and α′β′ KCs labeled. Scale bar, 10 μm (**H**) Sparseness of γ KC population responses is not affected by mAChR-A knockdown (p=0.76, two-way repeated-measures ANOVA). (**I**) Correlation between γ KC population responses to MCH and OCT is not affected by mAChR-A knockdown (p=0.32, t-test).

DOI: https://doi.org/10.7554/eLife.48264.015

The following source data is available for figure 4:

**Source data 1.** Source data for ***Figure 4***.
DOI: https://doi.org/10.7554/eLife.48264.016

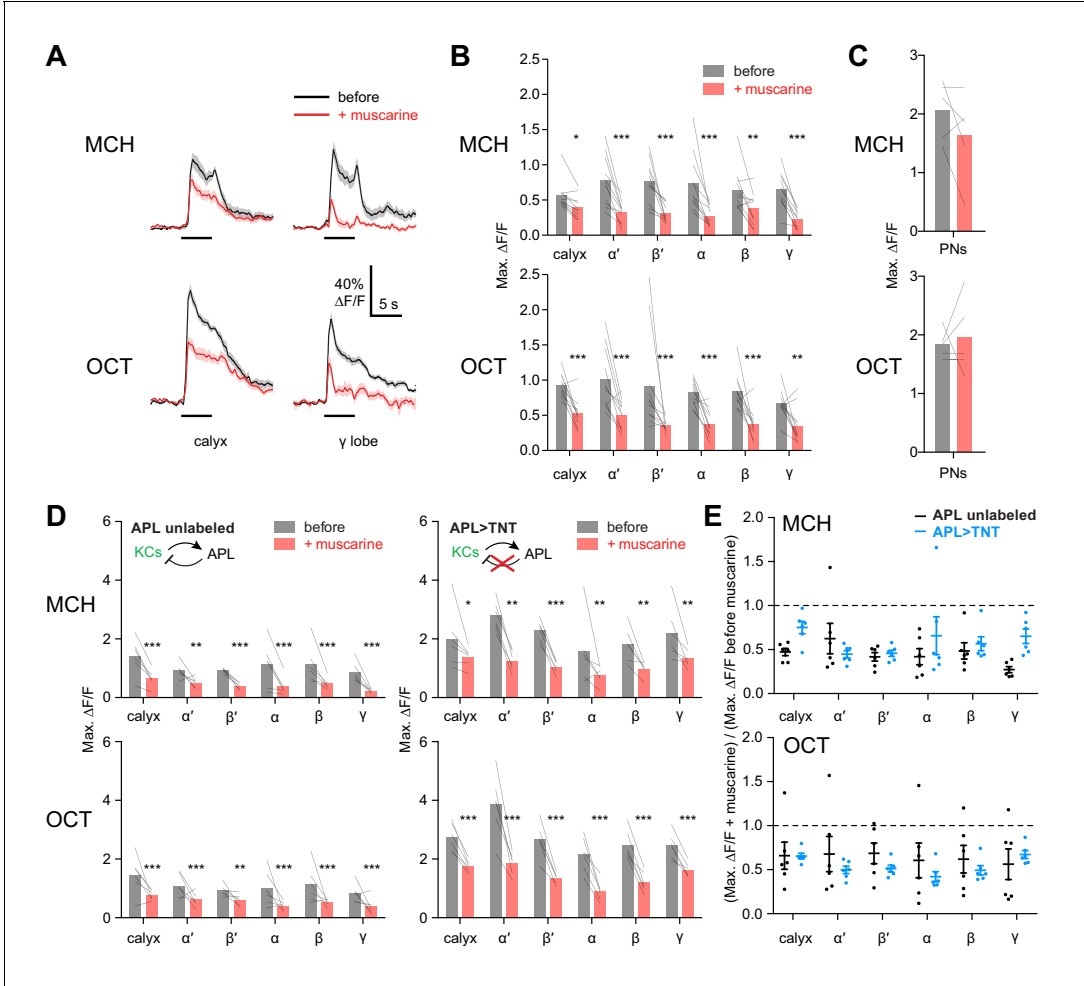

**Figure 5.** KC odor responses are decreased by muscarine. (A) Odor responses in the calyx and γ lobe of OK107-GAL4 > GCaMP6f flies, before (black) and after (red) adding 10 µM muscarine in the bath. Data are mean (solid line) ± SEM (shaded area); horizontal lines indicate the odor pulse. Traces for all lobes are shown in *Figure 5—figure supplement 1*. For detailed statistical analysis see *Supplementary file 1*. (B) Peak ΔF/F during the odor pulse before and after muscarine. n = 11 hemispheres from 6 flies. *p<0.05, **p<0.01, ***p<0.001 by two-way repeated measures ANOVA with Holm-Sidak multiple comparisons test. (C) Odor responses in PN axons in the calyx are not affected by 10 µM muscarine, in GH146-GAL4 > GCaMP6f flies (p>0.49, two-way repeated measures ANOVA, n = 5 flies). (D) Peak ΔF/F during the odor pulse before and after muscarine in control hemispheres where APL was unlabeled (left, n = 6 hemispheres from 6 flies) and hemispheres where APL expressed tetanus toxin (TNT) (right, n = 6 hemispheres from 5 flies). *p<0.05, **p<0.01, ***p<0.001 by two-way repeated measures ANOVA with Holm-Sidak multiple comparisons test. (E) (Response (peak ΔF/F during the odor pulse) after muscarine) / (response before muscarine), using data from (D). No significant differences were observed (p>0.05, two-way repeated measures ANOVA with Holm-Sidak multiple comparisons test).

DOI: https://doi.org/10.7554/eLife.48264.017

The following source data and figure supplement are available for figure 5:

**Source data 1.** Source data for *Figure 5*.

DOI: https://doi.org/10.7554/eLife.48264.019

**Figure supplement 1.** KC odor responses are decreased by muscarine — all traces.

DOI: https://doi.org/10.7554/eLife.48264.018

control hemispheres and hemispheres where APL synaptic output was blocked by tetanus toxin (*Figure 5D*), and the effect of muscarine was not significantly different between the two cases (*Figure 5E*). This result indicates that muscarine does not act solely by activating APL or by enhancing inhibition on KCs (e.g. increasing membrane localization of GABA_A receptors).

To test mAChR-A function even more acutely, we locally applied muscarine to the MB calyx by pressure ejection (*Figure 6*, *Figure 6—figure supplement 1*). Red dye included in the ejected solution confirmed that the muscarine remained in the calyx for several seconds but did not spread to

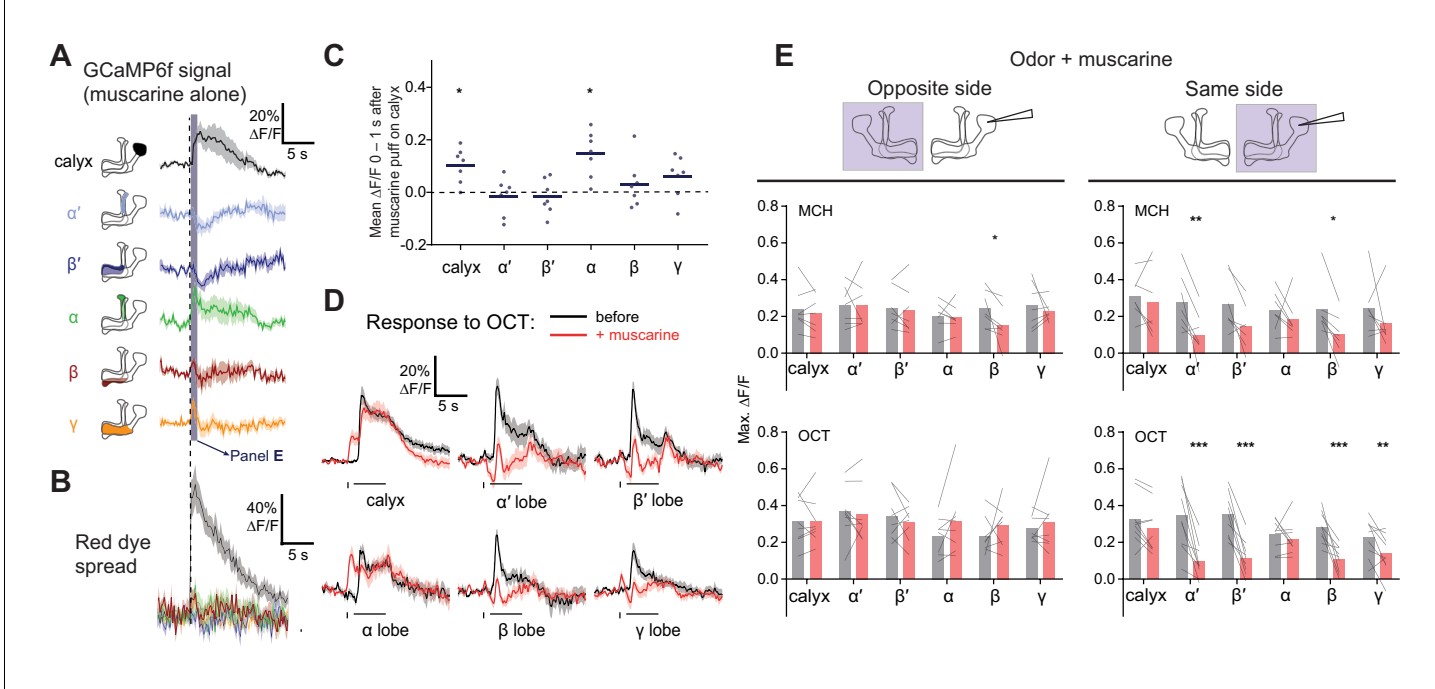

**Figure 6.** Local muscarine application to the calyx inhibits KC odor responses. (**A**) Left: Schematic of MB, showing color scheme for the different regions where responses are quantified. Right: Average ΔF/F GCaMP6f signal in different areas of the MB of OK107 > GCaMP6f flies in response to a 10 ms pulse of 20 mM muscarine on the calyx. Data are mean (solid line) ± SEM (shaded area). Dashed vertical line shows the timing of muscarine application. Shaded bar indicates time window used to quantify responses in panel C. n = 7 hemispheres (5 flies). (**B**) ΔF/F traces of red dye indicator, showing which MB regions the muscarine spread to. The traces follow the same color scheme and visuals as shown in panel A. (**C**) Scatter plot showing average ΔF/F of GCaMP6f signal of the different MB regions at time 0–1 s 10 ms pulse of 20 mM muscarine on the calyx, quantified from traces shown in (**A**). n as in (**A**). *p<0.05, one-sample t-test (different from 0), Bonferroni correction for multiple comparisons. (**D**) Average ΔF/F GCaMP6f signal of different areas of the MB during odor pulses of OCT (horizontal bar), before (black) and after (red) muscarine application on the calyx, 1 s before the odor pulse (vertical bar). Data are mean (solid line) ± SEM (shaded area). n: 7 hemispheres (5 flies). See *Figure 6—figure supplement 1* for all traces. (**E**) Line-bar plots showing paired peak ΔF/F GCaMP6f responses of the different MB regions during 5 s odor pulses of MCH or OCT, before (gray) and after (pink) muscarine application to the calyx, in the hemisphere where the muscarine was applied (same side, right) or the opposite (opposite side, left). Muscarine was applied 1 s before the odor pulse. Bars show mean value. n given as number of hemispheres (number of flies): Same side MCH 7 (6), OCT 9 (8), opposite side MCH 7 (5), OCT 8 (5). *p<0.05, **p<0.01, ***p<0.001 by two-way repeated measures ANOVA with Holm-Sidak multiple comparisons test.

DOI: https://doi.org/10.7554/eLife.48264.020

The following source data and figure supplement are available for figure 6:

**Source data 1.** Source data for *Figure 6*.

DOI: https://doi.org/10.7554/eLife.48264.022

**Figure supplement 1.** Local muscarine application to the calyx inhibits KC odor responses — all traces.

DOI: https://doi.org/10.7554/eLife.48264.021

the MB lobes (*Figure 6B*). Surprisingly, applying muscarine to the calyx in the absence of odor stimuli increased GCaMP6f signal in the calyx and α lobe, with small increases in the β and γ lobe that were not statistically significant (*Figure 6A,C*). It also decreased GCaMP6f signal in the α' and β' lobes around 1–2 s after application (*Figure 6A*), although this effect was also not statistically significant. The increased $Ca^{2+}$ in the calyx most likely did not reflect increased excitability, as applying muscarine to the calyx did not increase the calyx odor response (*Figure 6D,E*). If anything, it likely *de*creased the calyx odor response, because the $Ca^{2+}$ increase induced by muscarine alone (no odor) lasted ~6–7 s and thus would have continued into the odor pulse in the muscarine +odor condition. If the odor response was unaffected by muscarine, the muscarine-evoked and odor-evoked increases in GCaMP6f signal should have summed. Instead, the peak calyx ΔF/F during the odor pulse was the same before and after locally applying muscarine, suggesting that the specifically odor-evoked increase in GCaMP6f was decreased by muscarine.

Indeed, applying muscarine to the calyx suppressed odor responses in KC axons (*Figure 6D,E*). Although muscarine did not significantly affect peak ΔF/F during the odor in the α lobe, muscarine most likely did decrease α lobe odor responses, by the same logic as for calyx odor responses (see above). Given that calyx muscarine suppresses α′β′ axonal odor responses, the decrease in α′β′ KC GCaMP6f signal in the absence of odor likely reflects suppression of spontaneous action potentials (*Figure 6A,C*), as α′β′ KCs have the highest spontaneous spike rate out of the three subtypes (*Groschner et al., 2018*; *Turner et al., 2008*). The effect of muscarine on α′β′ KCs is consistent with single-cell transcriptome analyses showing that α′β′ KCs express mAChR-A, albeit at a lower level than αβ or γ KCs (*Figure 2—figure supplement 1*) (*Croset et al., 2018*; *Davie et al., 2018*). The increase in calyx $Ca^{2+}$ induced by muscarine alone (without odor) might reflect $Ca^{2+}$ release from internal stores triggered by $G_q$ signaling, which then inhibits KC excitability (thus smaller odor responses). Note that muscarine on the calyx is unlikely to reduce KC odor responses via presynaptic inhibition of PNs, because bath muscarine does not affect odor-evoked $Ca^{2+}$ influx in PNs in the calyx (*Figure 5C*), although we cannot rule out $Ca^{2+}$-independent inhibition.

## mAChR-A localized to the MB calyx can rescue learning in a mAChR-A hypomorphic mutant

We next asked where mAChR-A exerts its effect. To visualize the localization of mAChR-A, we created a new construct with mAChR-A tagged with FLAG on the C-terminus under UAS control. When we overexpressed FLAG-tagged mAChR-A in KCs using OK107-GAL4, we only observed anti-FLAG staining in the calyx (*Figure 7A*), suggesting that mAChR-A is localized to the calyx. To test whether the FLAG tag or overexpression might cause the mAChR-A to be mis-localized, we tested whether mb247-GAL4 > mAChR A-FLAG overexpression could rescue learning in a mAChR-A mutant background. The original MiMIC allele with a GFP insertion in the 5′ UTR intron of mAChR-A contains a stop cassette and polyadenylation signal, and indeed, it is a strongly hypomorphic allele: qPCR shows almost total lack of mAChR-A mRNA in the 'MiMIC-stop' allele (*Figure 7B*). Flies homozygous for the 'MiMIC-stop' allele are viable but show impaired learning, while learning is significantly improved by using mb247-GAL4 to overexpress mAChR-A-FLAG in αβ and γ KCs (*Figure 7C*), indicating that overexpressed mAChR-A-FLAG can support learning. These flies ('MiMIC-stop', mb247 >mAChR A-FLAG) also show anti-FLAG staining only in the calyx (*Figure 7—figure supplement 1*). These results suggest that mAChR-A exerts its effect on learning in KC dendrites, consistent with the effect of locally applying muscarine to KC dendrites.

## mAChR-A knockdown prevents training-induced depression of MBON odor responses

The finding that mAChR-A functions in KC dendrites raises the question of how mAChR-A can affect learning. While learning-associated plasticity in KC dendrites has been observed in honeybees, In *Drosophila*, olfactory associative memories are stored by weakening the synapses between KCs and output neurons that lead to the 'wrong' behavior. For example, aversive memory requires an output neuron downstream of γ KCs, called MBON-γ1pedc>α/β or MB-MVP2. MB-MVP2 leads to approach behavior (*Aso et al., 2014b*), and aversive conditioning reduces MB-MVP2's responses to the aversively-trained odor (*Hige et al., 2015*; *Perisse et al., 2016*). We tested whether knocking down mAChR-A would prevent this depression. We knocked down mAChR-A in KCs using OK107-GAL4 and UAS-mAChR-A-RNAi 1, and expressed GCaMP6f in MB-MVP2 using R12G04-LexA and lexAop-GCaMP6f (*Figure 8A*). We trained flies in the behavior apparatus and then imaged MB-MVP2 odor responses (3 hr after training to avoid cold-shock-sensitive memory). Because overall response amplitudes were variable across flies, for each fly we measured the ratio of the response to MCH (the trained odor) over the response to OCT (the untrained odor). Consistent with previous published results (*Hige et al., 2015*; *Perisse et al., 2016*), in control flies not expressing mAChR-A RNAi, the MCH/OCT ratio was substantially reduced in trained flies relative to mock-trained flies (*Figure 8B*). This was not because the OCT response increased, because there was no difference between trained and mock-trained flies in the ratio of the response to OCT over the response to isoamyl acetate, a 'reference' odor that was absent in the training protocol. This was also not because of any general decrease in odor responses, as shown by analyzing absolute response amplitudes to MCH, OCT and isoamyl acetate (*Figure 8—figure supplement 1*). In contrast, in flies expressing mAChR-A RNAi in

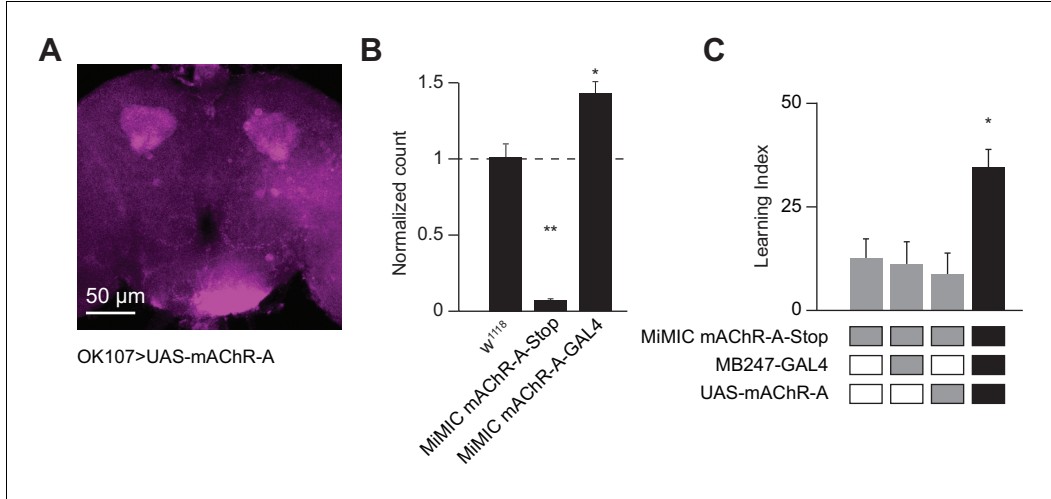

OK107>UAS-mAChR-A

**Figure 7.** Dendritic function of mAChR-A suffices to rescue learning in mAChR-A mutants. (**A**) mAChR-A-FLAG overexpressed in KCs by OK107-GAL4 appears in the calyx but not the lobes of the mushroom body. (**B**) Flies homozygous for the MiMIC mAChR-A-stop allele (which contains a stop cassette as part of the Minos gene-trap cassette in the 5'UTR) have virtually no mAChR-A mRNA. In contrast, flies with the MiMIC mAChR-A-GAL4 allele do not have reduced mAChR-A mRNA levels, because the stop cassette was replaced with GAL4 (indeed, their mAChR-A levels are slightly higher than the control). (mean ± SEM; n = 4 each with three technical replicates; \*\*p=0.0001; Welch ANOVA test with Dunnett's T3 multiple comparisons test). For detailed statistical analysis see *Supplementary file 1*. (**C**) Homozygous MiMIC mAChR-A-stop flies are defective in olfactory aversive learning, but learning is rescued by driving mAChR-A-FLAG in αβ and γ KCs by mb247-GAL4. n (left to right): 49, 70, 56, 47, \*p<0.05, Kruskal-Wallis test with Dunn's multiple comparisons test). For detailed statistical analysis see *Supplementary file 1*.

DOI: https://doi.org/10.7554/eLife.48264.023

The following source data and figure supplement are available for figure 7:

**Source data 1.** Source data for *Figure 7B*.
DOI: https://doi.org/10.7554/eLife.48264.025
**Source data 2.** Source data for *Figure 7C*.
DOI: https://doi.org/10.7554/eLife.48264.026
**Figure supplement 1.** Localization of mb247-GAL4 > mAChR-A-FLAG.
DOI: https://doi.org/10.7554/eLife.48264.024

KCs, the MCH/OCT ratio was the same between trained and mock-trained flies (*Figure 8B*), indicating that the mAChR-A knockdown impaired the learning-related depression of the KC to MB-MVP2 synapse. This result suggests that mAChR-A function in KC dendrites is necessary for learning-related synaptic plasticity in KC axons.

## Discussion

Here, we show that mAChR-A is required in γ KCs for aversive olfactory learning and short-term memory in adult *Drosophila*. Knocking down mAChR-A increases KC odor responses, while the mAChR-A agonist muscarine suppresses KC activity. Knocking down mAChR-A prevents aversive learning from reducing responses of the MB output neuron MB-MVP2 to the conditioned odor, suggesting that mAChR-A is required for the learning-related depression of KC->MBON synapses.

Why is mAChR-A only required for aversive learning in γ KCs, not αβ or α'β' KCs? Although our mAChR-A MiMIC gene trap agrees with single-cell transcriptome analysis that α'β' KCs express less mAChR-A than do γ and αβ KCs (*Croset et al., 2018*; *Davie et al., 2018*), transcriptome analysis indicates that α'β' KCs do express some mAChR-A (*Figure 2—figure supplement 1*). Moreover, γ and αβ KCs express similar levels of mAChR-A (*Crocker et al., 2016*). It may be that the RNAi knockdown is less efficient at affecting the physiology of αβ and α'β' KCs than γ KCs, whether because the knockdown is less efficient at reducing protein levels, or because αβ and α'β' KCs have

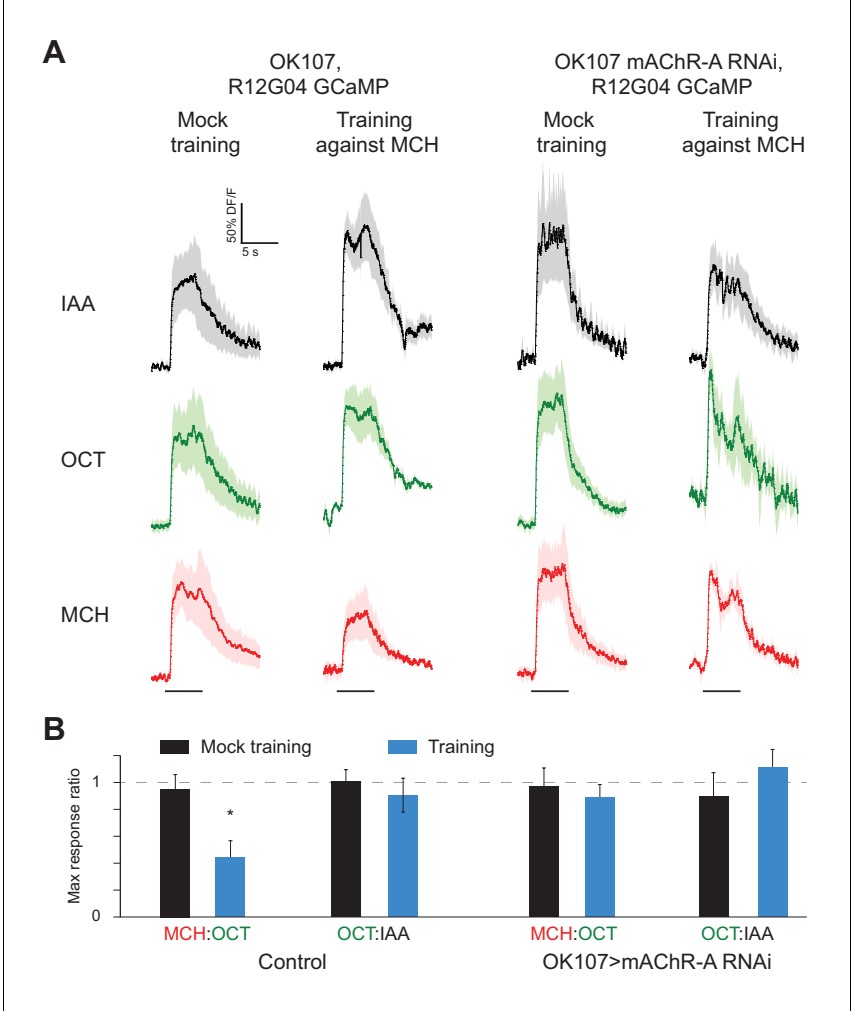

**Figure 8.** mAChR-A knockdown prevents aversive conditioning from decreasing the response to the trained odor in MB-MVP2. (**A**) Odor responses in MB-MVP2 to isoamyl acetate (IAA, not presented during training), OCT (not shocked during training) and MCH (shocked during training), in control (OK107-GAL4, R12G04-LexA > GCaMP6f, mb247-dsRed) and knockdown (OK107-GAL4 > mAChR A-RNAi 1, R12G04-LexA > GCaMP6f, mb247-dsRed) flies, with mock training (no shock) or training against MCH. Traces show mean (solid line) ± SEM (shaded area). (**B**) MCH:OCT or OCT:IAA ratios of peak ΔF/F values from (**A**). n = 5. *p<0.05, Mann-Whitney test. Power analysis shows that n = 5 would suffice to detect an effect as strong as the difference between training and mock training in the MCH:OCT ratio, with power 0.9. See *Figure 8—figure supplement 1* for absolute ΔF/F values.
DOI: https://doi.org/10.7554/eLife.48264.027

The following source data and figure supplement are available for figure 8:

**Source data 1.** Source data for *Figure 8* and *Figure 8—figure supplement 1*.
DOI: https://doi.org/10.7554/eLife.48264.029

**Figure supplement 1.** Diagram and additional data for *Figure 8* (mAChR-A knockdown prevents learning-associated depression of odor responses in MVP2).
DOI: https://doi.org/10.7554/eLife.48264.028

different intrinsic properties or a different function of mAChR-A such that 40% of normal mAChR-A levels is sufficient in αβ and α′β′ KCs but not γ KCs. This interpretation is supported by our finding that mAChR-A RNAi knockdown significantly increases odor responses only in the γ lobe, not the αβ or α′β′ lobes. Alternatively, γ, αβ and α′β′ KCs are thought to be important mainly for short-term memory, long-term memory, and memory consolidation, respectively (*Guven-Ozkan and Davis, 2014*; *Krashes et al., 2007*); as we only tested short-term memory, mAChR-A may carry out the same function in all KCs, but only its role in γ KCs is required for short-term (as opposed to long-

term) memory. Indeed, the key plasticity gene DopR1 is required in γ, not αβ or α′β′ KCs, for short-term memory (*Qin et al., 2012*). It may be that mAChR-A is required in non-γ KC types for other forms of memory besides short-term aversive memory, such as appetitive conditioning or other phases of memory like long-term memory. Our finding that mAChR-A is required in γ KCs for aversive short-term memory is consistent with our finding that mAChR-A knockdown in KCs disrupts training-induced depression of odor responses in MB-MVP2, an MBON postsynaptic to γ KCs required for aversive short-term memory (*Perisse et al., 2016*). However, the latter finding does not rule out the possibility that other MBONs postsynaptic to non-γ KCs may also be affected by mAChR-A knockdown in KCs.

mAChR-A seems to inhibit KC odor responses, because knocking down mAChR-A increases odor responses in the calyx and γ lobe, while activating mAChR-A with bath or local application of muscarine decreases KC odor responses. Some details differ between the genetic and pharmacological results. In particular, while mAChR-A knockdown mainly affects γ KCs, with other subtypes inconsistently affected, muscarine reduces responses in all KC subtypes. What explains these differences? mAChR-A might be weakly activated in physiological conditions, in which case gain of function would cause a stronger effect than loss of function. Similarly, pharmacological activation of mAChR-A is likely a more drastic manipulation than a 60% reduction of mAChR-A mRNA levels. Although we cannot entirely rule out network effects from muscarine application, the effect of muscarine does not stem from PNs or APL (*Figure 5C,D*) and locally applied muscarine would have little effect on neurons outside the mushroom body.

How does mAChR-A inhibit odor-evoked $Ca^{2+}$ influx in KCs? Given that mAChR-A signals through $G_q$ when expressed in CHO cells (*Ren et al., 2015*), that muscarinic $G_q$ signaling normally increases excitability in mammals (*Caulfield and Birdsall, 1998*), and that pan-neuronal artificial activation of $G_q$ signaling in *Drosophila* larvae increases overall excitability (*Becnel et al., 2013*), it may be surprising that mAChR-A inhibits KCs. However, $G_q$ signaling may exert different effects on different neurons in the fly brain, and some examples exist of inhibitory $G_q$ signaling by mammalian mAChRs. $M_1/M_3/M_5$ receptors acting via $G_q$ can inhibit voltage-dependent $Ca^{2+}$ channels (*Gamper et al., 2004*; *Kammermeier et al., 2000*; *Keum et al., 2014*; *Suh et al., 2010*), reduce voltage-gated Na +currents (*Cantrell et al., 1996*), or trigger surface transport of KCNQ channels (*Jiang et al., 2015*), thus increasing inhibitory $K^+$ currents. *Drosophila* mAChR-A may inhibit KCs through similar mechanisms.

What is the source of ACh which activates mAChR-A and modulates odor responses? In the calyx, cholinergic PNs are certainly a major source of ACh. However, KCs themselves are cholinergic (*Barnstedt et al., 2016*) and release neurotransmitter in both the calyx and lobes (*Christiansen et al., 2011*). KCs form synapses on each other in the calyx (*Zheng et al., 2018*), possibly allowing mAChR-A to mediate lateral inhibition, in conjunction with the lateral inhibition provided by the GABAergic APL neuron (*Lin et al., 2014*).

What function does mAChR-A serve in learning and memory? Our results indicate that mAChR-A knockdown prevents the learning-associated weakening of KC-MBON synapses, in particular for MBON-γ1pedc>α/β, aka MB-MVP2 (*Figure 7*). One potential explanation is that the increased odor-evoked $Ca^{2+}$ influx observed in knockdown flies increases synaptic release, which overrides the learning-associated synaptic depression. However, increased odor-evoked $Ca^{2+}$ influx per se is unlikely on its own to straightforwardly explain a learning defect, because other genetic manipulations that increase odor-evoked $Ca^{2+}$ influx in KCs either have no effect on, or even improve, olfactory learning. For example, knocking down GABA synthesis in the inhibitory APL neuron increases odor-evoked $Ca^{2+}$ influx in KCs (*Lei et al., 2013*; *Lin et al., 2014*) and improves olfactory learning (*Liu and Davis, 2009*).

The most intuitive explanation would be that mAChR-A acts at KC synaptic terminals in KC axons to help depress KC-MBON synapses. Yet overexpressed mAChR-A localizes to KC dendrites, not axons, and functionally rescues mAChR-A hypomorphic mutants, showing that dendritic mAChR-A suffices for its function in learning and memory. Does this show that mAChR-A has no role in KC axons? Our inability to detect GFP expressed from the mAChR-A MiMIC gene trap suggests that normally there may only be a small amount of mAChR-A in KCs. It may be that with mAChR-A-FLAG overexpression, the correct (undetectable) amount of mAChR-A is trafficked to and functions in axons, but due to a bottleneck in axonal transport, the excess tagged mAChR-A is trapped in KC dendrites. While our results do not rule out this possibility, a general bottleneck in axonal transport

seems unlikely as many overexpressed proteins are localized to KC axons (*Trunova et al., 2011*). We feel it is more parsimonious to take the dendritic localization of mAChR-A-FLAG at face value and infer that mAChR-A functions in KC dendrites.

How can mAChR-A in KC dendrites affect synaptic plasticity in KC axons? mAChR-A signaling might change the shape or duration of KC action potentials (*Allen and Burnstock, 1990*; *Ghamari-Langroudi and Bourque, 2004*), an effect that could potentially propagate to KC axon terminals (*Juusola et al., 2007*; *Shu et al., 2006*). Such changes in the action potential waveform may not be detected by calcium imaging, but could potentially affect a 'coincidence detector' in KC axons that detects when odor (i.e. KC activity) coincides with reward/punishment (i.e. dopamine). This coincidence detector is generally believed to be the $Ca^{2+}$-dependent adenylyl cyclase *rutabaga* (*Levin et al., 1992*). Changing the waveform of KC action potentials could potentially affect local dynamics of $Ca^{2+}$ influx near *rutabaga* molecules. In addition, *rutabaga* mutations do not abolish learning (mutants have ~40–50% of normal learning scores) (*Yildizoglu et al., 2015*), so there may be additional coincidence detection mechanisms affected by action potential waveforms. Testing this idea would require a better understanding of biochemical events underlying learning at KC synaptic terminals.

Alternatively, mAChR-A's effects on synaptic plasticity may not occur acutely. Although we ruled out purely developmental effects of mAChR-A through adult-only RNAi expression (*Figure 1E*), knocking out mAChR-A for several days in adulthood might still affect KC physiology in a not-entirely-acute way. For example, as with other G-protein-coupled receptors (*Wang and Zhuo, 2012*), muscarinic receptors can affect gene expression (*von der Kammer et al., 1998*), which could have wide-ranging effects on KC physiology, for example action potential waveform, expression of key genes required for synaptic plasticity, etc. Another intriguing possibility is suggested by an apparent paradox: both mAChR-A and the dopamine receptor Damb signal through $G_q$ (*Himmelreich et al., 2017*), but mAChR-A promotes learning while Damb promotes forgetting (*Berry et al., 2012*). How can $G_q$ mediate apparently opposite effects? Perhaps $G_q$ signaling aids both learning and forgetting by generally rendering synapses more labile. Indeed, although *damb* mutants retain memories for longer than wildtype, their initial learning is slightly impaired (*Berry et al., 2012*); *damb* mutant larvae are also impaired in aversive olfactory learning (*Selcho et al., 2009*). Although one study reports that knocking down $G_q$ in KCs did not impair initial memory (*Himmelreich et al., 2017*), the $G_q$ knockdown may not have been strong enough; also, that study shocked flies with 90 V shocks, which also gives normal learning in mAChR-A knockdown flies (*Figure 1—figure supplement 1*).

Such hypotheses posit that mAChR-A regulates synaptic plasticity 'competence' rather than participating directly in the plasticity mechanism itself. Why should synaptic plasticity competence be controlled by an activity-dependent mechanism? It is tempting to speculate that mAChR-A may allow a kind of metaplasticity (*Abraham, 2008*) in which exposure to odors (hence activation of mAChR-A in KCs) makes flies' learning mechanisms more sensitive. Indeed, mAChR-A is required for learning with moderate (50 V) shocks, not severe (90 V) shocks. Future studies may further clarify how muscarinic signaling contributes to olfactory learning.

# Materials and methods

## Key resources table

| Reagent type (species) or resource | Designation | Source or reference | Identifiers | Additional information |
|---|---|---|---|---|
| Gene (*Drosophila melanogaster*) | mAChR-A | | FLYB: FBgn0000037 | Also known as: mAChR, mAcR-60C, DM1, Acr60C, CG4356 |
| Genetic reagent (*D. melanogaster*) | *MiMIC mAChR-A-stop* | (*Venken et al., 2011*) PMID 21985007 | BDSC:59216 | *mAChR-A^{MI13848}* |
| Genetic reagent (*D. melanogaster*) | *UAS-GCaMP6f (attP40)* | (*Chen et al., 2013*) PMID 23868258 | BDSC:42747 | |

*Continued on next page*

*Continued*

| Reagent type (species) or resource | Designation | Source or reference | Identifiers | Additional information |
|---|---|---|---|---|
| Genetic reagent (*D. melanogaster*) | *UAS-GCaMP6f (VK00005)* | (*Chen et al., 2013*) PMID 23868258 | BDSC:52869 | |
| Genetic reagent (*D. melanogaster*) | *lexAop-GCaMP6f* | (*Barnstedt et al., 2016*) PMID 26948892 | | Gift from S. Waddell |
| Genetic reagent (*D. melanogaster*) | *UAS-mAChR-A RNAi 1* | Bloomington *Drosophila* Stock Center | BDSC:27571 | TRiP.JF02725 |
| Genetic reagent (*D. melanogaster*) | *UAS-mAChR-A RNAi 2* | Vienna *Drosophila* Resource Center | VDRC:101407 | |
| Genetic reagent (*D. melanogaster*) | *UAS-Dcr-2* | Bloomington *Drosophila* Stock Center | BDSC:24651 | |
| Genetic reagent (*D. melanogaster*) | *lexAop-GAL80* | Bloomington *Drosophila* Stock Center | BDSC:32216 | |
| Genetic reagent (*D. melanogaster*) | *tub-GAL80$^{ts}$* | (*McGuire et al., 2003*) PMID 14657498 | BDSC:7108 | |
| Genetic reagent (*D. melanogaster*) | *mb247-dsRed* | (*Riemensperger et al., 2005*) PMID 16271874 | FLYB:FBtp0022384 | |
| Genetic reagent (*D. melanogaster*) | *GH146-GAL4* | (*Stocker et al., 1997*) PMID 9110257 | BDSC:30026 | |
| Genetic reagent (*D. melanogaster*) | *OK107-GAL4* | (*Connolly et al., 1996*) PMID 8953046 | BDSC:854 | |
| Genetic reagent (*D. melanogaster*) | *c305a-GAL4* | (*Krashes et al., 2007*) PMID 17196534 | BDSC:30829 | |
| Genetic reagent (*D. melanogaster*) | *mb247-GAL4* | (*Zars et al., 2000*) PMID 10784450 | BDSC:50742 | |
| Genetic reagent (*D. melanogaster*) | *R44E04-LexA* | (*Jenett et al., 2012*) PMID 23063364 | BDSC:52736 | Gift from A. Thum |
| Genetic reagent (*D. melanogaster*) | *R45H04-LexA* | (*Bräcker et al., 2013*) PMID 23770186 | FLYB:FBti0155893 | Gift from A. Thum |
| Genetic reagent (*D. melanogaster*) | *R12G04-LexA* | (*Jenett et al., 2012*) PMID 23063364 | BDSC:52448 | |
| Genetic reagent (*D. melanogaster*) | *elav-GAL4* | (*Lin and Goodman, 1994*) PMID 7917288 | BDSC:458 | |
| Genetic reagent (*D. melanogaster*) | *NP2631-GAL4* | (*Lin et al., 2014*; *Tanaka et al., 2008*) PMID 24561998, 18395827 | Kyoto Stock Center 104266 | |
| Genetic reagent (*D. melanogaster*) | *GH146-FLP* | (*Hong et al., 2009*; *Lin et al., 2014*) PMID 19915565, 24561998 | FLYB:FBtp0053491 | |
| Genetic reagent (*D. melanogaster*) | *tub-FRT-GAL80-FRT* | (*Gordon and Scott, 2009*; *Lin et al., 2014*) PMID 19217375, 24561998 | BDSC:38880 | |
| Genetic reagent (*D. melanogaster*) | *UAS-TNT* | (*Lin et al., 2014*; *Sweeney et al., 1995*) PMID 24561998, 7857643 | FLYB:FBtp0001264 | |
| Genetic reagent (*D. melanogaster*) | *UAS-mCherry-CAAX* | (*Kakihara et al., 2008*; *Lin et al., 2014*) PMID 18083504, 24561998 | FLYB:FBtp0041366 | |
| Genetic reagent (*D. melanogaster*) | *mb247-LexA* | (*Lin et al., 2014*; *Pitman et al., 2011*) PMID 24561998 | FLYB:FBtp0070099 | |

*Continued on next page*

*Continued*

| Reagent type (species) or resource | Designation | Source or reference | Identifiers | Additional information |
|---|---|---|---|---|
| Genetic reagent (*D. melanogaster*) | *20xUAS-6xGFP* | (*Shearin et al., 2014*) PMID 24451596 | BDSC:52266 | |
| Genetic reagent (*D. melanogaster*) | *UAS-mCD8-GFP* | (*Lee et al., 1999*) PMID 10457015 | BDSC:5130 | |
| Antibody | nc82 (mouse monoclonal) | Developmental Studies Hybridoma Bank | nc82 | (1:50, supernatant or 1:200, concentrate) |
| Antibody | FLAG (mouse monoclonal M2) | Sigma-Aldrich | F3165 | (1:250) |
| Antibody | Goat anti-mouse secondary Alexa 647 | Abcam | ab150115 | (1:500) |
| Antibody | Goat anti-mouse secondary Alexa 546 | Thermo Fisher | A11018 | (1:1000) |

## Fly strains

Fly strains (see below) were raised on cornmeal agar under a 12 hr light/12 hr dark cycle and studied 1–10 days post-eclosion. Strains were cultivated at 25°C unless they expressed temperature-sensitive gene products (GAL80$^{ts}$); in these cases, the experimental animals and all relevant controls were grown at 23°C. To de-repress the expression of RNAi with GAL80$^{ts}$, experimental and control animals were incubated at 31°C for 7 days. Subsequent behavioral experiments were performed at 25°C.

Experimental animals carried transgenes over Canton-S chromosomes where possible to minimize genetic differences between strains. Details of fly strains are given in the Key Resources Table.

UAS-mAChR-A-FLAG plasmid was generated by Gibson assembly of fragments using the NEBuilder HiFi Master Mix (NEB). Fragments were created by PCR using Phusion High-Fidelity DNA Polymerase (NEB). The full-length mAChR-A cDNA was purchased from GenScript (clone ID OFa11160). The vector was pTWF-attB, a gift from Prof. Oren Schuldiner (*Yaniv et al., 2012*). This vector consists of a FLAG tag in the C-terminal of the inserted gene and an attB site for site-specific integration of the transgene. PCR and Gibson assembly were carried out following the manufacturer's recommendations with the following primers:

For mAChR-A: tgggaattatcgacaagtttgtacaaaaaagcaggctATGGAGCCGGTCATGAGTC and cactttgtacaagaaagctgggtaATTGTAGACGCCGCGTAC

For pTWF-AttB: aaagctgggtaCTTGTACAAAGTGGTGAGCTCC and agcctgctttttgtacAAACTTGTCGATAATTCCC

Transgenes were injected into the attP2 landing site using φC31 integration (by BestGene).

## Quantitative real-time PCR

Total RNA was extracted by EZ-RNA II Total RNA Isolation kit (Biological Industries, Israel) from 30 adult heads for each biological replicate. cDNA was generated from 1 µg total RNA with the High-Capacity cDNA Reverse Transcription Kit with RNase Inhibitor (Applied Biosystems). Real-time quantitative PCR was carried with TaqMan Fast Advanced Master Mix (Applied Biosystems) and run in technical triplicates on a StepOne Plus Real-Time PCR System (Applied Biosystems). Taqman assays were Dm01820303_g1 for mAChR-A and Dm02151962_g1 for EF1 (Ef1alpha100E, ThermoFisher). The expression levels obtained for mAChR-A were normalized to those of the housekeeping gene EF1. The fold change for mAChR-A was subsequently calculated by comparing to the normalized value of either ELAV-gal4 parent (for RNAi experiments) or w$^{1118}$ flies (for MIMiC experiments).

## Behavioral analysis

Behavioral experiments were performed in a custom-built, fully automated apparatus (*Claridge-Chang et al., 2009*; *Lin et al., 2014*; *Parnas et al., 2013*). Single flies were housed in clear

polycarbonate chambers (length 50 mm, width 5 mm, height 1.3 mm) with printed circuit boards (PCBs) at both floors and ceilings. Solid-state relays (Panasonic AQV253) connected the PCBs to a 50 V source.

Air flow was controlled with mass flow controllers (CMOSens PerformanceLine, Sensirion). A carrier flow (2.7 l/min) was combined with an odor stream (0.3 l/min) obtained by circulating the air flow through vials filled with a liquid odorant. Odors were prepared at 10 fold dilution in mineral oil. Therefore, liquid dilution and mixing carrier and odor stimulus stream resulted in a final 100 fold dilution of odors. Fresh odors were prepared daily.

The 3 l/min total flow (carrier and odor stimulus) was split between 20 chambers resulting in a flow rate of 0.15 l/min per half chamber. Two identical odor delivery systems delivered odors independently to each half of the chamber. Air or odor streams from the two halves of the chamber converged at a central choice zone. The 20 chambers were stacked in two columns each containing 10 chambers and were backlit by 940 nm LEDs (Vishay TSAL6400). Images were obtained by a MAKO CMOS camera (Allied Vision Technologies) equipped with a Computar M0814-MP2 lens. The apparatus was operated in a temperature-controlled incubator (Panasonic MIR-154) maintained at 25°C.

A virtual instrument written in LabVIEW 7.1 (National Instruments) extracted fly position data from video images and controlled the delivery of odors and electric shocks. Data were analyzed in MATLAB 2015b (The MathWorks) and Prism 6 (GraphPad).

A fly's preference was calculated as the percentage of time that it spent on one side of the chamber. Training and odor avoidance protocols were as depicted in *Figure 1*. The naïve avoidance index was calculated as (preference for left side when it contains air) – (preference for left side when it contains odor). During training, MCH was paired with 12 equally spaced 1.25 s electric shocks at 50 V (*Tully and Quinn, 1985*). The learning index was calculated as (preference for MCH before training) – (preference for MCH after training). Flies were excluded from analysis if they entered the choice zone fewer than four times during odor presentation.

## Functional imaging

Brains were imaged by two-photon laser-scanning microscopy (*Ng et al., 2002*; *Wang et al., 2003*). Cuticle and trachea in a window overlying the required area were removed, and the exposed brain was superfused with carbogenated solution (95% $O_2$, 5% $CO_2$) containing 103 mM NaCl, 3 mM KCl, 5 mM trehalose, 10 mM glucose, 26 mM $NaHCO_3$, 1 mM $NaH_2PO_4$, 3 mM $CaCl_2$, 4 mM $MgCl_2$, 5 mM N-Tris (TES), pH 7.3. Odors at $10^{-1}$ dilution were delivered by switching mass-flow controlled carrier and stimulus streams (Sensirion) via software controlled solenoid valves (The Lee Company). Flow rates at the exit port of the odor tube were 0.5 or 0.8 l/min.

Fluorescence was excited by a Ti-Sapphire laser centered at 910 nm, attenuated by a Pockels cell (Conoptics) and coupled to a galvo-resonant scanner. Excitation light was focussed by a 20X, 1.0 NA objective (Olympus XLUMPLFLN20XW), and emitted photons were detected by GaAsP photomultiplier tubes (Hamamatsu Photonics, H10770PA-40SEL), whose currents were amplified and transferred to the imaging computer. Two imaging systems were used, #1 for *Figures 3–6* except **5C**, and #2 for *Figure 5C* and *Figure 7*, which differed in the following components: laser (1: Mai Tai eHP DS, 70 fs pulses; 2: Mai Tai HP DS, 100 fs pulses; both from Spectra-Physics); microscope (1: Movable Objective Microscope; 2: DF-Scope installed on an Olympus BX51WI microscope; both from Sutter); amplifier for PMT currents (1: Thorlabs TIA-60; 2: Hamamatsu HC-130-INV); software (1: ScanImage 5; 2: MScan 2.3.01). Volume imaging on System 1 was performed using a piezo objective stage (nPFocus400, nPoint). Muscarine was applied locally by pressure ejection from borosilicate patch pipettes (resistance ~10 MOhm; capillary inner diameter 0.86 mm, outer diameter 1.5 mm; concentration in pipette 20 mM; pressure 12.5 psi) using a Picospritzer III (Parker). A red dye was added to the pipette to visualize the ejected fluid (SeTau-647, SETA BioMedicals) (*Podgorski et al., 2012*).

Movies were motion-corrected in X-Y using the moco ImageJ plugin (*Dubbs et al., 2016*), with pre-processing to collapse volume movies in Z and to smooth the image with a Gaussian filter (standard deviation = 4 pixels; the displacements generated from the smoothed movie were then applied to the original, unsmoothed movie), and motion-corrected in Z by maximizing the pixel-by-pixel correlation between each volume and the average volume across time points. ΔF/F, activity maps, sparseness and inter-odor correlation were calculated as in *Lin et al. (2014)*. Briefly, movies were smoothed with a 5-pixel-square Gaussian filter (standard deviation 2). Baseline fluorescence was

taken as the average fluorescence during the pre-stimulus period. Frames with sudden, large axial movements were discarded by correlating each frame to the baseline image and discarding it if the correlation fell below a threshold value, which was manually selected for each brain by noting the constant high correlation value when the brain was stationary and sudden drops in correlation when the brain moved. ΔF/F was calculated for each pixel as the difference between mean fluorescence during the stimulus period vs. the baseline fluorescence (ΔF), divided by the baseline fluorescence. For pixels where ΔF did not exceed two times the standard deviation over time of that pixel's intensity during the pre-stimulus period, the pixel was considered non-responsive. We excluded non-responsive flies and flies whose motion could not be corrected.

Inter-odor correlations were calculated by first aligning the activity maps of each odor response by maximizing the inter-odor correlations of baseline fluorescence, and then converting image matrices of the activity maps of each odor response into linear vectors and calculating the Pearson correlation coefficients between each 'odor vector'. A threshold for baseline fluorescence was applied as a mask to the activity map to exclude pixels with no baseline GCaMP6f signal. Population sparseness was calculated for activity maps using the following equation (*Vinje and Gallant, 2000*; *Willmore and Tolhurst, 2001*):

$$S_P = \frac{1}{1 - \frac{1}{N}} \left( 1 - \frac{\left( \sum_{i=1}^{N} \frac{r_i}{N} \right)^2}{\sum_{i=1}^{N} \frac{r_i^2}{N}} \right)$$

## Structural imaging

Brain dissections, fixation, and immunostaining were performed as described (*Pitman et al., 2011*; *Wu and Luo, 2006*). To visualize native GFP fluorescence, dissected brains were fixed in 4% (w/v) paraformaldehyde in PBS (1.86 mM $NaH_2PO_4$, 8.41 mM $Na_2HPO_4$, 175 mM NaCl) and fixed for 20 min at room temperature. Samples were washed for 3 × 20 min in PBS containing 0.3% (v/v) Triton-X-100 (PBT). The neuropil was counterstained with nc82 (DSHB) or monoclonal anti-FLAG M2 antibody (F3165, Sigma) and goat anti-mouse Alexa 647 or Alexa 546. Primary antisera were applied for 1–2 days and secondary antisera for 1–2 days in PBT at 4°C, followed by embedding in Vectashield. Images were collected on a Leica TCS SP5, SP8, or Nikon A1 confocal microscope and processed in ImageJ.

APL expression of tetanus toxin was scored by widefield imaging of mCherry. mCherry expression in APL was distinguished from 3XP3-driven dsRed from the GH146-FLP transgene by using separate filter cubes for dsRed (49004, Chroma: 545/25 excitation; 565 dichroic; 605/70 emission) and mCherry (LED-mCherry-A-000, Semrock: 578/21 excitation; 596 dichroic; 641/75 emission).

## Statistics

Statistical analyses were carried out in GraphPad Prism as described in figure legends and *Supplementary file 1*. In general, no statistical methods were used to predetermine sample sizes, but where conclusions were drawn from the absence of a statistically significant difference, a power analysis was carried out in G*Power to confirm that the sample size provided sufficient power to detect an effect of the expected size. The experimenter was blind to which hemispheres had APL neurons expressing tetanus toxin before post-experiment dissection (*Figure 5*) but not otherwise.

## Acknowledgements

We thank Vincent Croset, Christoph Treiber and Scott Waddell for sharing mAChR-A expression data before publication. We thank Oren Schuldiner, Andreas Thum, Scott Waddell, the Bloomington Stock Center, the Vienna *Drosophila* RNAi Center, and the Kyoto *Drosophila* Genetic Resource center for plasmids and fly strains. We thank Lily Bolsover for technical assistance. We thank Anton Nikolaev for comments on the manuscript. This work was supported by the European Research Council (676844, MP; 639489, AL) and the Deutsche Forschungsgemeinschaft (HU 2474/1-1, WH).

## Additional information

### Funding

| Funder | Grant reference number | Author |
|---|---|---|
| European Commission | 676844 | Moshe Parnas |
| European Commission | 639489 | Andrew C Lin |
| Deutsche Forschungsgemeinschaft | HU 2474/1-1 | Wolf Huetteroth |

The funders had no role in study design, data collection and interpretation, or the decision to submit the work for publication.

### Author contributions

Noa Bielopolski, Anthi A Apostolopoulou, Formal analysis, Investigation, Visualization, Methodology, Writing—review and editing; Hoger Amin, Formal analysis, Investigation, Visualization, Methodology, Writing—original draft, Writing—review and editing; Eyal Rozenfeld, Software, Formal analysis, Investigation, Visualization, Methodology, Writing—review and editing; Hadas Lerner, Formal analysis, Investigation, Writing—review and editing; Wolf Huetteroth, Investigation, Visualization, Writing—review and editing; Andrew C Lin, Conceptualization, Software, Formal analysis, Supervision, Funding acquisition, Investigation, Visualization, Methodology, Writing—original draft, Writing—review and editing; Moshe Parnas, Conceptualization, Software, Formal analysis, Supervision, Funding acquisition, Investigation, Visualization, Methodology, Writing—original draft, Writing—review and editing, Initiated the project

### Author ORCIDs

Hoger Amin https://orcid.org/0000-0002-7884-4815
Anthi A Apostolopoulou https://orcid.org/0000-0002-8174-4372
Eyal Rozenfeld https://orcid.org/0000-0001-5316-0495
Wolf Huetteroth https://orcid.org/0000-0003-3421-9935
Andrew C Lin https://orcid.org/0000-0001-6310-9765
Moshe Parnas https://orcid.org/0000-0001-9726-1511

### Decision letter and Author response

Decision letter https://doi.org/10.7554/eLife.48264.034
Author response https://doi.org/10.7554/eLife.48264.035

## Additional files

### Supplementary files

• Supplementary file 1. Details of statistical analysis.
DOI: https://doi.org/10.7554/eLife.48264.030

• Supplementary file 2. Detailed genotypes used in this study.
DOI: https://doi.org/10.7554/eLife.48264.031

• Transparent reporting form
DOI: https://doi.org/10.7554/eLife.48264.032

### Data availability

All data generated or analysed during this study are included in the manuscript and supporting files. Source data files have been provided for Figures 1-8, Figure 1—figure supplement 1, and Figure 8—figure supplement 1.

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
