## [Decision Letter]

[Editors’ note: a previous version of this study was rejected after peer review, but the authors submitted for reconsideration. The first decision letter after peer review is shown below.]

Thank you for submitting your work entitled "Inhibitory muscarinic acetylcholine receptors enhance aversive olfactory conditioning in adult *Drosophila*" for consideration by *eLife*. Your article has been reviewed by three peer reviewers, including Mani Ramaswami as the Reviewing Editor and Reviewer #1, and the evaluation has been overseen by a Reviewing Editor and a Senior Editor.

Our decision has been reached after consultation between the reviewers. Based on these discussions and the individual reviews below, we regret to inform you that your work will not be considered further for publication in *eLife*. However, we do feel the work could be made acceptable but only after the essential changes noted below are handled experimentally.

All the reviewers note the interest in the core observation that the *Drosophila* mAChR mediates inhibition in the mushroom body as well that it is required in Kenyon cells for aversive associative memory. However, the mechanism by which it functions, in terms of the receptor's localisation at Kenyon cell presynaptic terminals and role in mediating signalling required for learning-associated depression of γ Kenyon cell – MBON synapses is not sufficiently well established. In addition to responding constructively to the various specific comments listed in the appended reviews, the reviewers felt that would be crucially important to: (a) demonstrate (potentially using lines expressing genomically-tagged mAChRs) the physical presence of mAChR in γ Kenyon cell presynapses; and (b) reanalyse data for the crucial Figure 7D in a way shows changes to MCH/OCT and IAA induced responses separately, in order to clarify that there is no depression to either the shock paired or shock unpaired odorant. (The current Figure 7D could potentially be explained if there were depression observed in both cases).

*Reviewer #1:*

Bielopolsky and coauthors analyse the expression muscarinic AcRs in the *Drosophila* mushroom body as well as in their function in the physiology and plasticity of Kenyon cells that involved in encoding associative memory. This is interesting as the role of metabotropic ACh signalling has not been previously analysed at this level of resolution, in neurons and synapses that encode memory. The major conclusion of the paper is that mAchR is required in γ Kenyon cells to mediate synaptic depression that underlies reduced response to odor that has been associated with an aversive stimulus. This conclusion is well justified, based on several well-done experiments that point to the cell-autonomous function of mAChRs in Kenyon cells, their physiological role in reducing odor-evoked response, and their function both for physiological depression and associative memory formation. It is also broadly interesting mAChR function in *Drosophila* may prove to be anagolous to the role of mGluR in mammalian synapses. The paper is well written and clear.

Other, less clearly resolved but interesting aspects of the paper involve fundamental aspects of mAChR physiology in the calyx and in different classes of Kenyon cell axons, as revealed by RNAi based knockdown experiments and direct muscarine application. Muscarine application increases basal calcium-levels in calyx, but reduces odor evoked responses in axons may be explained as the authors proposed, but there could be alternative explanations.

Essential revisions:

1) Figure 7 shows normal levels of memory – which superficially contradicts Figure 2. Better and more clear labelling should be used to show that this is only true when there are much higher levels of aversive stimulation. This is also confusing because the imaging experiments shown in subsequent panels of the same figure use "normal" and lower levels of aversive stimulation.

2) The observations Figure 6 (subsection “KC odor responses are decreased by an mAChR agonist”) seem to be overinterpreted. I think the data are fine to include, but the differences in OCT responses vs. MCH responses in the different classes of Kenyon cell axons etc. are not particularly compelling and may simply be at the level of experimental and biological noise. I suggest focusing on the main, robust and interpretable thread of these observations, and to present the other data observations that will take follow up work to understand completely.

3) I also found the robust effect of mAChR in α′β′ cells to be confusing, given its apparent weak expression in this class of Kenyon cell. Can this be explained – perhaps the expression is not perfectly revealed by the MiMIC conversion line?

4) In subsection “mAChR-A is required for olfactory learning in γ KCs, not αβ or α′β′ KCs”, "γ neurons project to the horizontal lobes only, while the axons of αβ and α′β′ neurons bifurcate to form the α an α′ portions of the vertical lobes and the β and β′ portions of the horizontal lobes" is a bit confusing.

*Reviewer #2:*

In this manuscript, the authors knock down a muscarinic acetylcholine receptor (mAChR-A) first, in all Kenyon cells (KCs) and then in subsets, while reading out flies' ability to form odor-shock associations as well as neuronal calcium responses. They conclude that mAChR-As mediate a spatially localized inhibition of odor responses in the dendritic as well as the axonal compartments of Kenyon cells and that this inhibition is needed in γ Kenyon cells, specifically in the horizontal lobe for learning.

A major failing is the use of imprecise RNAi based methods to knockdown mAChR-A, the efficacy of which in blocking behavior depends on the drivers used (see essential revisions 1). Another issue is the way the Calcium imaging data was analyzed. The authors use imprecise analyses and then report a negative result (see essential revisions 1 and 2).

I feel the manuscript has convincingly shown that mAChRs mediate cholinergic inhibition and that this may affect learning in some way. This is an interesting, initial result. The additional claims about how it affects learning and that it is necessary in γ KCs lobes are not justified by the data and analyses. Significant re-analysis and possibly some new experiments might be needed in order to draw any mechanistic conclusions. How does the mAChR-A mediated inhibition influence plasticity? This question has not been answered here.

Essential revisions:

1) γ KC specificity claim is not justified:

The RNAi knock downs depended on the driver lines used (subsection “mAChR-A is required for olfactory learning in γ KCs, not αβ or α′β′ KCs”). The authors use different RNAi driver in the behavior (Figure 2) and imaging experiments (Figure 3). In fact, the significant reduction in response for OCT not seen in Figure 3, appears in Figure 4E, reinforcing the notion that the differences being detected here fluctuate depending on the strength of the manipulation and might not reflect real physiological differences.

The signal to noise of the Ca response trace would depend on the size of the ROI being averaged. More careful analysis and/or experiments would be needed to conclude that γ KCs have larger increases in response sizes than other cell types. The authors should either image and analyze single KC somata or at least, draw ROIs that average a fixed number of pixels from a single plane in the volume (averaging in z should also be controlled for).

2) Claims of lobe specific effects of localized muscarine application are not justified:

In Figure 6, the authors claim to measure Ca-responses to either muscarine alone or muscarine plus odor in the various lobes and compare the effect of local muscarine application. When muscarine was applied in the horizontal lobes (Figures 6C,D), the spread of muscarine into the various lobes could be non-uniform. The regions of interest over which ∆F/F values were calculated (as drawn in Figure 6A) would probably include different ratios of muscarine-applied and muscarine un-applied neuropil. Ca-fluorescence should only be extracted from regions of the lobes over-lapping with muscarine (red dye) spread. This should be straightforward to re-analyze. The lack of a clear and significant difference in most if not all the lobes in Figure 6G might also be a result of this effect.

*Reviewer #3:*

Bielopolsky et al. report learning experiments in *Drosophila melanogaster*. They use an RNAi approach to knockdown a metabotropic acetylcholine receptor in the mushroom body, and this affects aversive associative olfactory learning. Downregulation of this receptor in γ-type Kenyon cells, but not other Kenyon cell types, is sufficient to impair learning. Odor-evoked calcium activity increases mainly in the Kenyon cell input region of the mushroom body (calyx), but also to some degree in axonal regions and not for both odorants used. Application of the receptor agonist muscarine also slightly decreased odor-evoked calcium activity in Kenyon cells when applied to the axons and slightly increased baseline calcium. However, application of muscarine to the calyx increased baseline calcium more pronounced and decreased odor-evoked calcium influx. Local injections of muscarine into lobes cause a slight decrease of calcium in γ lobes which leads them to conclude that mAChR-A might be present in axons and contribute to learning-induced synaptic plasticity. The data I do not find convincing because the restriction of the injected drug is only confirmed by dye application. Injection of the drug into the calyx causes effects in the α lobe. It might be possible that injection into the in γ lobe also affects calycal input. The argument that increasing the strength of the US compensates for the learning deficit induced by the RNAi construct I do not find convincing because one could conclude that a modified CS (changed through the RNAi expression) requires a stronger US to be learned. Also, the conclusion that "odor coding" is not affected by the mAchR-A knockdown I find difficult given that odor-evoked calcium is significantly increased. The plasticity observed in the MB-MVP2 neuron 3 hours after training and the absence of this plasticity when the receptor is downregulated is a very indirect hint that the receptor might be present on axons. Overall, the finding that a muscarinic acetylcholine receptor is required in γ-type Kenyon cells for learning but not odor avoidance is very nice, and I would like to congratulate the authors for that interesting result. The conclusion that this receptor is also localized to axons and has a role in presynaptic plasticity is not convincing. Localization studies of the receptor would be required.

[Editors’ note: what now follows is the decision letter after the authors submitted for further consideration.]

Thank you for resubmitting your work entitled "Inhibitory muscarinic acetylcholine receptors enhance aversive olfactory learning in adult *Drosophila*" for further consideration at *eLife*. Your revised article has been favorably evaluated by K VijayRaghavan (Senior Editor), Mani Ramaswami as Reviewing Editor, and two reviewers.

The manuscript has been greatly improved by careful and constructive revisions but there is one remaining issue that needs to be addressed before acceptance.

We would like you to please consider the comments below (reproduced verbatim) and respond appropriately with revisions if you agree, or with a brief rebuttal if you do not, and to submit a final revised manuscript that will then be acceptable for publication.

1) The authors emphasize differential effects on subsets of KCs when the data do not justify these claims. The claim of differential effects on KC subtypes may detract from the clear communication of the central observations.

a) The authors claim in Figure 2 with the driver MB247 Gal4 that only γ KC based learning involves AChR-A, by receptor knockdown. The behavioral readout here is aversive, differential odor conditioning with a specific CS-US pairing protocol. Had the authors used appetitive reinforcement or spaced training protocols with more repeats, they might have seen behavioral effects with other KC subtypes as well. Furthermore, they use a different driver, OK107 Gal4 for knockdown in Figure 3 to claim that odor responses are significantly increased by receptor knockdown only in γ KCs. The use of different drivers to draw a parallel between behavior and neuronal activity is not strictly correct.

b) The main claim made in Figure 6 is that muscarine application in the calyx differentially affects KC subtypes. This claim is based on the observation that muscarine application in the calyx depressed odor responses in all lobes except the α lobe. It is hard to explain how a manipulation in the calyx differentially affects only the α lobe axonal branch of α/β KCs. The odor-response traces for this same α lobe in Figure 6D rise when muscarine is applied, before odor delivery. It is unclear if the experiment is clean enough to support the claim being made. Note also that the γ KCs, which are described as the main targets of the effect of mAChR-A in Figure 2 and Figure 3 behave similarly to other KC subtypes in this figure.

c) Figure 8 shows that mAChR-A knockdown prevents learning-related plasticity in a γ KC compartment. This does not show that other KCs and KC compartments are not similarly affected by mAChR-A.

d) I would instead argue that Figure 5 indicates that odor responses in all KC subtypes are affected by mAChRs in a similar, important, surprising manner; consistent with the behavior and MBON imaging observations.

---

## [Author Response]

[Editors’ note: the author responses to the first round of peer review follow.]

In addition to responding constructively to the various specific comments listed in the appended reviews, the reviewers felt that would be crucially important to: (a) demonstrate (potentially using lines expressing genomically-tagged mAChRs) the physical presence of mAChR in γ Kenyon cell presynapses;

We tried to use CRISPR to generate an mAChR-A-FRT-stop-FRT-V5 allele that would allow endogenous mAChR-A to be conditionally tagged with V5 in cells expressing FLP. However, due to various problems, we have not managed to obtain the correct flies.

In parallel, we took an alternative approach: we overexpressed mAChR-A-FLAG in KCs and observed anti-FLAG immunostaining in the calyx, not the lobes. This

suggests that mAChR-A is localized to KC dendrites. This overexpressed construct can also rescue learning in a mAChR-A hypomorph background, showing that dendritically localized mAChR-A functions in learning (i.e., the dendritic FLAG signal isn’t an artifact like misfolded mAChR-A stuck in the ER). See subsection “mAChR-A localized to the MB calyx can rescue learning in a mAChR-A hypomorphic mutant” and new Figure 7. Even though this doesn’t use the suggested genomically-tagged mAChR-A, we feel this is strong evidence that mAChR-A functions in KC dendrites.

Thus, based on new evidence, we changed our original conclusion. Although we

cannot rule out the possibility that a low, undetectable level of mAChR-A functions in KC axons (see seventh paragraph of the Discussion section) we feel the most parsimonious explanation is that mAChR-A indeed functions in KC dendrites.

… and (b) reanalyse data for the crucial Figure 7D in a way shows changes to MCH/OCT and IAA induced responses separately, in order to clarify that there is no depression to either the shock paired or shock unpaired odorant. (The current Figure 7D could potentially be explained if there were depression observed in both cases).

We reanalyzed the data used for Figure 7 in a new Figure 8—figure supplement 1 (note previous Figure 7 is now Figure 8) and demonstrate that there is no general depression in odor responses.

Reviewer #1:[…]Essential revisions:1) Figure 7 shows normal levels of memory – which superficially contradicts figure 2. Better and more clear labelling should be used to show that this is only true when there are much higher levels of aversive stimulation. This is also confusing because the imaging experiments shown in subsequent panels of the same figure use "normal" and lower levels of aversive stimulation.

We have moved the confusing panel Figure 7A to Figure 1—figure supplement 1A (with added labeling to note “90 V shock”) (see also response to reviewer #3, second point).

2) The observations Figure 6 (subsection “KC odor responses are decreased by an mAChR agonist”) seem to be overinterpreted. I think the data are fine to include, but the differences in OCT responses vs. MCH responses in the different classes of Kenyon cell axons etc. are not particularly compelling and may simply be at the level of experimental and biological noise. I suggest focusing on the main, robust and interpretable thread of these observations, and to present the other data observations that will take follow up work to understand completely.

In light of all three reviewers’ concerns about the localized application of muscarine to the lobe, we removed these results. We kept the localized application of muscarine to the calyx, but we focused our interpretation on the temporal rather than spatial specificity of application (i.e., muscarine affects GCaMP signal on the scale of < 1 s). See subsection “KC odor responses are decreased by an mAChR agonist”.

3) I also found the robust effect of mAChR in α′β′ cells to be confusing, given its apparent weak expression in this class of Kenyon cell. Can this be explained – perhaps the expression is not perfectly revealed by the MiMIC conversion line?

We added this sentence to the Results following the discussion of the MiMIC conversion line: “However, mAChR-A is still clearly present in α′β′ KCs’ transcriptomes, suggesting that mAChR-A-MiMIC-GAL4 may not reveal all neurons that express mAChR-A.” See subsection “KC odor responses are decreased by an mAChR agonist”.

We also added this sentence to the Results discussing muscarine application: “The effect of muscarine on α′β′ KCs is consistent with single-cell transcripome analyses showing that α′β′ KCs express mAChR-A, albeit at a lower level than αβ or γ KCs (Figure 2—figure supplement 1).” See subsection “KC odor responses are decreased by an mAChR agonist”.

4) Subsection “mAChR-A is required for olfactory learning in γ KCs, not αβ or α′β′ KCs”: "γ neurons project to the horizontal lobes only, while the axons of αβ and α′β′ neurons bifurcate to form the α an α′ portions of the vertical lobes and the β and β′ portions of the horizontal lobes" is a bit confusing.

We now write: “Kenyon cells are subdivided into three main classes according to their innervation of the horizontal and vertical lobes of the MB: γ neurons send axons only to the γ lobe of the horizontal lobes, while the axons of αβ and α′β′ neurons bifurcate and go to both the vertical and horizontal lobes (αβ axons make up the α lobe of the vertical lobe and β lobe of the horizontal lobe, while α′β′ axons make up the α′ lobe of the vertical lobe and β′ portion of the horizontal lobe).” See subsection “mAChR-A is required for olfactory learning in γ KCs, not αβ or α′β′ KCs”.

Reviewer #2:[…]Essential revisions:1) γ KC specificity claim is not justified:The RNAi knock downs depended on the driver lines used (subsection “mAChR-A is required for olfactory learning in γ KCs, not αβ or α′β′ KCs”). The authors use different RNAi driver in the behavior (Figure 2) and imaging experiments (Figure 3).

This comment may be due to a misunderstanding. Actually, we used the same RNAi lines and GAL4 drivers in both behavior and imaging. In particular, we used both RNAi 1 and RNAi 2 for behavior (RNAi 1 in Figure 1, Figure 2; RNAi 2 in Figure 1) and for imaging (RNAi 1 in Figure 4; RNAi 2 in Figure 3). We used the same GAL4 drivers for both behavior and imaging: OK107 to label all KCs (Figure 1 behavior; Figure 3 imaging), and mb247-GAL4, R44E04-LexA>lexAop-GAL80 (Figure 2 behavior, Figure 4 imaging). It’s true that in Figure 2 we also tested some other drivers on behavior, but as they either caused no behavior defect or were less specific than the γ-specific driver, there was no need to test them on imaging. We apologize for our text being unclear, and we have added the following text to stress this point: “(this driver and RNAi combination was also used for behavior in Figure 1C)” (subsection “mAChR-A suppresses odor responses in γ KCs”) and “the same driver and RNAi combination used in the behavioral experiments in Figure 2B” (subsection “mAChR-A suppresses odor responses in γ KCs”).

In fact, the significant reduction in response for OCT not seen in Figure 3, appears in Figure 4E, reinforcing the notion that the differences being detected here fluctuate depending on the strength of the manipulation and might not reflect real physiological differences.

Unfortunately, we don’t understand this comment. First, we found that mAChR-A

RNAi causes an increase in KC odor responses, not a reduction. Second, comparing Figure 3 and Figure 4E, the response to OCT is increased in both the calyx and γ lobe in both figures.

The signal to noise of the Ca response trace would depend on the size of the ROI being averaged. More careful analysis and/or experiments would be needed to conclude that γ KCs have larger increases in response sizes than other cell types. The authors should either image and analyze single KC somata or at least, draw ROIs that average a fixed number of pixels from a single plane in the volume (averaging in z should also be controlled for).

It is true that the signal-to-noise ratio (SNR) of the measured GCaMP signal differed between the various regions of the mushroom body (depending on both ROI size and z-depth, as deeper z-planes give less signal due to light scattering). However, the SNR was high enough that these differences do not affect the statistical power of our comparisons. We measured the SNR as the reciprocal of the coefficient of variation (CoV, or standard deviation divided by mean) of GCaMP signal over time during the pre-stimulus period (2 s before the odor). Because ∆F/F = (F-F0)/F0 = F/F0 – 1 (so F/F0 = ∆F/F + 1), and F0 = mean signal during the pre-stimulus period, the CoV of fluorescence during the pre-stimulus interval equals the standard deviation of ∆F/F during the pre-stimulus interval.

The CoV was in general very low (0.02 – 0.05, corresponding to a SNR of 20 – 50). Differences in SNR in this range would not substantially affect the chances of detecting an effect size as large as what we observed in the γ lobe (effect size, Cohen’s *d* = 1.3 for OCT, 1.8 for MCH). We simulated 2 groups of 20 random samples (n=20 was the smallest sample size out of the αβ and α′β′ lobes) where the effect size of the difference between the 2 groups was 1.3. Each sample had a ‘ground truth’ value, from which we sampled 3 ‘time points’ that were subject to noise with SNR from 1–50 (we sampled 3 time points because the peak of the odor response almost always occurred between 1–2 s after odor onset, and our frame rate was ~3 Hz). The maximum of these 3 time points was taken as the measured ‘peak odor response’. We ran 1000 simulations, ran t-tests on the simulated data, and counted how many gave a p-value < 0.0125 (a Holm-Bonferroni correction for the 4 mushroom body regions that did not consistently show significant differences between control and mAChR-A-RNAi flies) – this fraction is the statistical power for detecting a difference in the non-γ lobes with effect size 1.3. The power was constant at ~0.91 for SNR above 12, whereas our SNRs were in the range 20–50. Thus, in our data, the SNR was high enough that differences in SNR between lobes would not affect the statistical power to detect an effect as large as what we observed in the γ lobe. This analysis now appears in new Figure 3—figure supplement1, referred to in subsection “mAChR-A suppresses odor responses in γ KCs”.

Still, to accommodate the reviewer’s concern, we have changed the framing of the relevant paragraph to “we next asked how odor responses in αβ, α′β′ and γ KCs are affected by mAChR-A knockdown”, and the concluding sentence to “However, we do not rule out the possibility that mAChR-A knockdown also affects αβ and α′β′ odor responses in a way that does not affect short-term memory, especially as αβ and α′β′ odor responses were somewhat, though non-significantly, increased.” See subsection “mAChR-A suppresses odor responses in γ KCs”.

2) Claims of lobe specific effects of localized muscarine application are not justified:In Figure 6, the authors claim to measure Ca-responses to either muscarine alone or muscarine plus odor in the various lobes and compare the effect of local muscarine application. When muscarine was applied in the horizontal lobes (Figure 6C,D), the spread of muscarine into the various lobes could be non-uniform. The regions of interest over which ∆F/F values were calculated (as drawn in Figure 6A) would probably include different ratios of muscarine-applied and muscarine un-applied neuropil. Ca-fluorescence should only be extracted from regions of the lobes over-lapping with muscarine (red dye) spread. This should be straightforward to re-analyze. The lack of a clear and significant difference in most if not all the lobes in Figure 6G might also be a result of this effect.

In light of all three reviewers’ concerns about the localized application of muscarine to the lobe, we removed these results. We kept the localized application of muscarine to the calyx, but we focused our interpretation on the temporal rather than spatial specificity of application (i.e., muscarine affects GCaMP signal on the scale of < 1 s). See subsection “KC odor responses are decreased by an mAChR agonist”.

Reviewer #3:Bielopolsky et al. report learning experiments in Drosophila melanogaster. They use an RNAi approach to knockdown a metabotropic acetylcholine receptor in the mushroom body, and this affects aversive associative olfactory learning. Downregulation of this receptor in γ-type Kenyon cells, but not other Kenyon cell types, is sufficient to impair learning. Odor-evoked calcium activity increases mainly in the Kenyon cell input region of the mushroom body (calyx), but also to some degree in axonal regions and not for both odorants used. Application of the receptor agonist muscarine also slightly decreased odor-evoked calcium activity in Kenyon cells when applied to the axons and slightly increased baseline calcium. However, application of muscarine to the calyx increased baseline calcium more pronounced and decreased odor-evoked calcium influx. Local injections of muscarine into lobes cause a slight decrease of calcium in γ lobes which leads them to conclude that mAChR-A might be present in axons and contribute to learning-induced synaptic plasticity.The data I do not find convincing because the restriction of the injected drug is only confirmed by dye application. Injection of the drug into the calyx causes effects in the α lobe. It might be possible that injection into the in γ lobe also affects calycal input.

In light of all three reviewers’ concerns about the localized application of muscarine to the lobe, we removed these results. We kept the localized application of muscarine to the calyx, but we focused our interpretation on the temporal rather than spatial specificity of application (i.e., muscarine affects GCaMP signal on the scale of < 1 s). (Subsection “KC odor responses are decreased by an mAChR agonist”).

But we would like to clarify that the effect of calyx-applied muscarine on the α lobe does not necessarily mean that the drug spread farther than the dye indicator; rather, it could mean that an effect on KC dendrites altered KC spiking, which altered GCaMP signals in KC axons.

The argument that increasing the strength of the US compensates for the learning deficit induced by the RNAi construct I do not find convincing because one could conclude that a modified CS (changed through the RNAi expression) requires a stronger US to be learned.

We thank the reviewer for this comment. We now place this panel in the supplemental material (Figure 1—figure supplement 1A) and present it as a side note, without drawing any conclusions as to whether the RNAi affects the CS, US, or CS-US integration. We also discuss the concept raised by the reviewer in more depth in the Discussion section.

Also, the conclusion that "odor coding" is not affected by the mAchR-A knockdown I find difficult given that odor-evoked calcium is significantly increased.

As part of removing the previous point about the weaker US, we have also removed the reference to “odor coding”.

The plasticity observed in the MB-MVP2 neuron 3 hours after training and the absence of this plasticity when the receptor is downregulated is a very indirect hint that the receptor might be present on axons. Overall, the finding that a muscarinic acetylcholine receptor is required in γ-type Kenyon cells for learning but not odor avoidance is very nice, and I would like to congratulate the authors for that interesting result. The conclusion that this receptor is also localized to axons and has a role in presynaptic plasticity is not convincing. Localization studies of the receptor would be required.

See discussion of localization studies in response to the summary comments.

[Editors' note: the author responses to the re-review follow.]

We would like you to please consider the comments below (reproduced verbatim) and respond appropriately with revisions if you agree, or with a brief rebuttal if you do not, and to submit a final revised manuscript that will then be acceptable for publication.1) The authors emphasize differential effects on subsets of KCs when the data do not justify these claims. The claim of differential effects on KC subtypes may detract from the clear communication of the central observations.a) The authors claim in Figure 2 with the driver MB247 Gal4 that only γ KC based learning involves AChR-A, by receptor knockdown. The behavioral readout here is aversive, differential odor conditioning with a specific CS-US pairing protocol. Had the authors used appetitive reinforcement or spaced training protocols with more repeats, they might have seen behavioral effects with other KC subtypes as well.

We agree, and have added a sentence in the Discussion section: “It may be that mAChR-A is required in non-γ KC types for other forms of memory besides short-term aversive memory, e.g., appetitive conditioning or other phases of memory like long-term memory.”

Furthermore, they use a different driver, OK107 Gal4 for knockdown in Figure 3 to claim that odor responses are significantly increased by receptor knockdown only in γ KCs. The use of different drivers to draw a parallel between behavior and neuronal activity is not strictly correct.

We have added this caveat to the Results section: “However, note that we do not exclude the possibility that αβ- or α′β′-specific (as opposed to pan-KC) knockdown of mAChR-A might significantly increase αβ or α′β′ KC odor responses.” See subsection “mAChR-A suppresses odor responses in γ KCs”.

1B) The main claim made in Figure 6 is that muscarine application in the calyx differentially affects KC subtypes.

We agree that the figure title was too strongly stated. We have changed the title of Figure 6 and its supplement to “Local muscarine application to the calyx inhibits KC odor responses”.

This claim is based on the observation that muscarine application in the calyx depressed odor responses in all lobes except the α lobe. It is hard to explain how a manipulation in the calyx differentially affects only the α lobe axonal branch of α/β KCs. The odor-response traces for this same α lobe in Figure 6D rise when muscarine is applied, before odor delivery. It is unclear if the experiment is clean enough to support the claim being made.

We added the sentence: “Although muscarine did not significantly affect peak ∆F/F during the odor in the α lobe, muscarine most likely did decrease α lobe odor responses, by the same logic as for calyx odor responses (see above).” See subsection “KC odor responses are decreased by an mAChR agonist”.

We also note that the β lobe also shows a small bump in response to muscarine alone in Figure 6B, so at least the α and β branches show qualitatively similar responses (although we agree that it’s puzzling why the α lobe shows a more sustained increase in GcaMP6f signal compared to the β lobe). We added the phrase “with small increases in the β and γ lobe that were not statistically significant” (subsection “KC odor responses are decreased by an mAChR agonist”).

Note also that the γ KCs, which are described as the main targets of the effect of mAChR-A in Figure 2 and Figure 3 behave similarly to other KC subtypes in this figure.

We agree and we feel that we already address this point by discussing the different results for mAChR-A RNAi knockdown vs. muscarine application in the Discussion section, starting with “while mAChR-A knockdown mainly affects γ KCs, with other subtypes inconsistently affected, muscarine reduces responses in all KC subtypes.”.

c) Figure 8 shows that mAChR-A knockdown prevents learning-related plasticity in a γ KC compartment. This does not show that other KCs and KC compartments are not similarly affected by mAChR-A.

We agree, and added two sentences to the Discussion section: “Our finding that mAChR-A is required in γ KCs for aversive short-term memory is consistent with our finding that mAChR-A knockdown in KCs disrupts training-induced depression of odor responses in MB-MVP2, an MBON postsynaptic to γ KCs required for aversive short-term memory (Perisse et al., 2016). However, the latter finding does not rule out the possibility that other MBONs postsynaptic to non-γ KCs may also be affected by mAChR-A knockdown in KCs.”.

d) I would instead argue that Figure 5 indicates that odor responses in all KC subtypes are affected by mAChRs in a similar, important, surprising manner; consistent with the behavior and MBON imaging observations.

We agree and think that this point is conveyed by the above changes together with existing discussion.